# Divergent DNA methylation dynamics in marsupial and eutherian embryos

Bryony J. Leeke[1,2,3,9 ✉], Wazeer Varsally[1,9], Sugako Ogushi[1], Jasmin Zohren[1], Sergio Menchero[1], Aurélien Courtois[1], Daniel M. Snell[1,4], Aurélie Teissandier[5], Obah Ojarikre[1], Shantha K. Mahadevaiah[1], Fanny Decarpentrie[6], Rebecca J. Oakey[7], John L. VandeBerg[8] & James M. A. Turner[1 ✉]

Based on seminal work in placental species (eutherians)[1–10], a paradigm of mammalian development has emerged wherein the genome-wide erasure of parental DNA methylation is required for embryogenesis. Whether such DNA methylation reprogramming is, in fact, conserved in other mammals is unknown. Here, to resolve this point, we generated base-resolution DNA methylation maps in gametes, embryos and adult tissues of a marsupial, the opossum *Monodelphis domestica*, revealing variations from the eutherian-derived model. The difference in DNA methylation level between oocytes and sperm is less pronounced than that in eutherians. Furthermore, unlike the genome of eutherians, that of the opossum remains hypermethylated during the cleavage stages. In the blastocyst, DNA demethylation is transient and modest in the epiblast. However, it is sustained in the trophectoderm, suggesting an evolutionarily conserved function for DNA hypomethylation in the mammalian placenta. Furthermore, unlike that in eutherians, the inactive X chromosome becomes globally DNA hypomethylated during embryogenesis. We identify gamete differentially methylated regions that exhibit distinct fates in the embryo, with some transient, and others retained and that represent candidate imprinted loci. We also reveal a possible mechanism for imprinted X inactivation, through maternal DNA methylation of the Xist-like noncoding RNA *RSX*[11]. We conclude that the evolutionarily divergent eutherians and marsupials use DNA demethylation differently during embryogenesis.

Marsupials diverged from eutherians 160 million years ago, and provide unique insight into mammalian embryology[12]. Relative to that of eutherians, formation of the placenta occurs late, and implantation is transient. By contrast, pups are born early and complete development outside the uterus. Like eutherians, marsupials undergo X chromosome inactivation (XCI). However, they lack the eutherian XCI-initiating noncoding RNA *Xist*. Instead, the alternative noncoding RNA, *RSX*, is implicated in marsupial XCI[11]. Furthermore, in eutherians, XCI in the soma is random, but in marsupials, it is imprinted, with the paternal X always chosen for silencing[13].

Epigenomic profiling has not been performed in preimplantation marsupial embryos. DNA methylation, which in eutherians is globally reprogrammed and linked to key events in embryo development, is of particular interest[14]. In eutherians, the paternal genome undergoes active demethylation after fertilization. Subsequently, the paternal and maternal genomes undergo passive demethylation, resulting in global hypomethylation in the early blastocyst[1–6]. Global erasure of DNA methylation is not observed in non-mammalian vertebrates[15,16], and is thought to serve a mammal-specific function (for example, to permit early embryonic genome activation (EGA), erasure of paternal

methylation and epimutations, formation of the trophectoderm, establishment of pluripotency or expression of transposons regulating embryo development)[7–10]. These embryonic milestones occur over a more protracted period in the marsupial, making it a useful alternative model to investigate the role of DNA demethylation in mammalian embryogenesis. Marsupials possess the core vertebrate DNA methylation machinery, including de novo methyltransferase genes (*DNMT3A* and *DNMT3B*), *DNMT3L*, *TET1–3*, *UHRF1* and two maintenance methyltransferase genes (*DNMT1A* and *DNMT1B*)[17]. However, the status of DNA methylation in gametes and during preimplantation development has not been explored.

## Cleavage embryos remain hypermethylated

To examine global methylation during marsupial embryogenesis, we generated low-input bisulfite sequencing (BS-seq) libraries from opossum sperm, oocytes and male and female preimplantation embryos, collected daily from embryonic day (E) 1.5 to E7.5, as well as triplicate adult somatic tissues representing the three germ layers (Fig. 1a and Supplementary Table 1). Our embryo series included the time points

[1]Sex Chromosome Biology Laboratory, The Francis Crick Institute, London, UK. [2]MRC Laboratory of Medical Sciences, London, UK. [3]Institute of Clinical Sciences, Imperial College London, London, UK. [4]Advanced Sequencing Facility, The Francis Crick Institute, London, UK. [5]INSERM U934, CNRS UMR3215, Institut Curie, PSL Research University, Paris, France. [6]Comparative Medicine, Novartis, Basel, Switzerland. [7]Department of Medical and Molecular Genetics, King's College London, London, UK. [8]Division of Human Genetics and South Texas Diabetes and Obesity Institute, School of Medicine, The University of Texas Rio Grande Valley, Brownsville, TX, USA. [9]These authors contributed equally: Bryony J. Leeke, Wazeer Varsally. ✉e-mail: b.leeke@lms.mrc.ac.uk; james.turner@crick.ac.uk

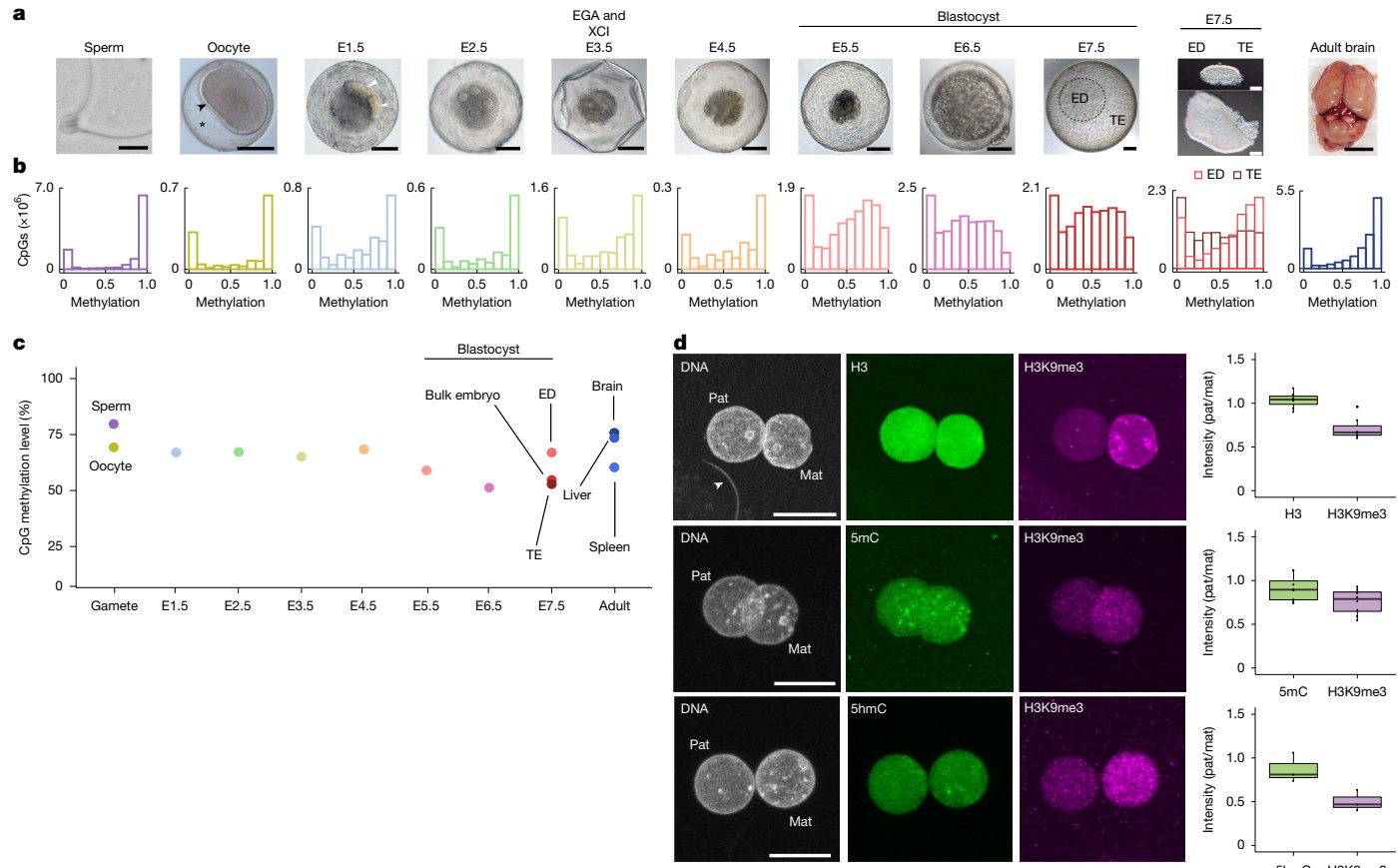

**Fig. 1 | DNA methylation dynamics in opossum embryos. a**, Light microscopy images of gametes, E1.5–E7.5 opossum embryos and brain. ED, embryonic disc; TE, trophectoderm. Scale bars, 10 μm (sperm), 100 μm (embryos) and 0.5 cm (brain). The black arrowhead marks the zona pellucida; the asterisk marks the mucoid coat; the white arrowheads mark blastomeres. Sample numbers are provided in Supplementary Table 1. **b**, Histograms of DNA methylation distribution at CpG sites captured at ≥5× coverage (note different scales on y axes). **c**, Mean DNA methylation level across developmental time. **d**, Micrographs: pronuclear-stage embryos (29.5 h post coitum) immunostained for histone H3

($n = 7$), 5mC ($n = 8$) or 5hmC ($n = 3$) and co-immunostained for H3K9me3. The two pronuclei (Pat, paternal; Mat, maternal) are of equivalent size (unlike in mice), but the paternal pronucleus is often marked by proximity to the sperm tail (arrowhead in first panel). Scale bars, 20 μm. Plots: quantification of the data. The centre line indicates the median, the bounds indicate the first and third quartiles, and the whiskers indicate 1.5× interquartile range (IQR). The more granular appearance of H3K9me3 in the middle and bottom panel is due to acid treatment of these samples, which is required for 5mC and 5hmC visualization.

of EGA (E3.5), XCI (E3.5), blastocyst formation (E5.5) and lineage divergence to form the trophectoderm and epiblast (EPI; E6.5 and E7.5; Supplementary Table 1). Control BS-seq on mice recapitulated published results[3], with sperm and brain exhibiting hypermethylation and a bimodal methylation pattern, and blastocysts showed global hypomethylation (mean methylation 76% in sperm, 71% in brain and 16% in blastocysts; Extended Data Fig. 1a).

In our opossum experiments, we obtained between 47 million and 337 million reads per library, with mapping rates between 15.7% and 71% (Supplementary Table 1). Principal component analysis separated sperm from other samples, with further separation by time point along principal component 2 (Extended Data Fig. 1b). After in silico pooling per time point, we captured between 49% and 91% of CpGs at 1× coverage, and 4.2% and 83% of CpG sites at 5× coverage (Extended Data Fig. 1c and Supplementary Table 1). As genomic coverage did not reach saturation in lower-input time points, we examined the proportion of genomic regions captured in each sample, and found that they were broadly similar between time points (Extended Data Fig. 1d). We therefore interpreted the global methylation distributions of each time point as representative of the overall genome.

Examination of the opossum methylation data revealed similarities to and differences from eutherians (Fig. 1b,c and Extended Data Fig. 2a). In opossums, like eutherians[3–6], the sperm genome was hypermethylated

relative to the oocyte. However, in opossums, the magnitude of the difference was smaller (mean methylation 77% in sperm and 65% in oocyte). As a result, the methylation level in oocytes was similar to that in somatic tissues (mean methylation 72%, 69% and 55% in brain, liver and spleen, respectively). Furthermore, following fertilization in the opossum, the genome remained relatively hypermethylated (mean methylation 62.5%). The E1.5 methylome was more similar to that of the oocyte, suggesting that some sites in the sperm are demethylated and reprogrammed to an oocyte-like state (Fig. 1b,c and Extended Data Fig. 1b). Nevertheless, the high levels of methylation in the early embryo demonstrated that the global loss of DNA methylation typical of eutherians did not occur in the opossum. To support this conclusion, we immunostained for 5-methylcytosine (5mC) and its derivative 5-hydroxymethylcytosine (5hmC) in opossum zygotes (Fig. 1d). In eutherians, demethylation of the paternal genome after fertilization results in lower 5mC and higher 5hmC levels in the paternal than in the maternal pronucleus[1,2,18,19]. We reproduced these findings in mice (Extended Data Fig. 2b). In the opossum, the paternal and maternal pronucleus contained equal levels of histone H3 but could be distinguished by histone 3 Lys9 tri-methylation (H3K9me3) staining, the level of which was lower in the paternal pronucleus (see Fig. 1d caption for details). In contrast to those of eutherians, 5mC and 5hmC levels in the opossum were similar between the paternal and maternal pronucleus.

DNA methylation dynamics in the paternal pronucleus therefore differ between opossums and eutherians.

We next tracked DNA methylation levels during cleavage and blastocyst formation. The hypermethylated state persisted up to and including E4.5, when the mean methylation was 64% (Fig. 1b,c). Subsequently, in the blastocyst, methylation initially decreased to 54% at E5.5 and 45% at E6.5, before rising to 49% at E7.5. Differential methylation testing of CpG sites between paired samples revealed that more than 90% of captured sites did not undergo significant changes in methylation level (Extended Data Fig. 2c). The stability in methylation was observed at all specific genomic features, including genes, intergenic regions and transposons (Extended Data Figs. 2d and 3a). In the last case, we nevertheless observed expression of multiple transposable element families at EGA (Extended Data Fig. 3b). This may be because expression derives from individual subfamily loci for which we did not capture sufficient BS-seq coverage to reveal locus-specific patterns. Alternatively, expression may be triggered independently of DNA demethylation. As observed in eutherians[20], promoters and CpG islands (CGIs) were largely hypomethylated across developmental time (Extended Data Fig. 2d). We conclude that, in contrast to the case in eutherians, in opposums, EGA and the cleavage divisions take place in the context of a hypermethylated genome.

## Divergent DNA methylation in EPI and trophectoderm

In the opossum, lineage divergence initiates at E6.5. To establish whether the decrease in methylation levels revealed by bulk analysis affected all cells or specific lineages, we performed BS-seq on isolated E7.5 embryonic disc (comprising EPI and primitive endoderm) and trophectoderm. The level of methylation was higher in the embryonic disc (63%) than in the trophectoderm (47%; Fig. 1b,c). This observation was consistent with the results of previous methylation-sensitive enzyme assays[21]. One explanation for this finding is that DNA methylation is retained in the EPI but reduced in the trophectoderm. An alternative is that methylation is reduced in both cell types and is then rapidly re-established in the EPI. To distinguish these possibilities, we performed single-cell multi-omics on embryos collected before lineage divergence (E5.5) and after lineage divergence (E6.5 and E7.5; Fig. 2a). As embryos transitioned from E5.5 to E6.5, a decrease in methylation was observed in both the EPI and the trophectoderm. Subsequently, at E7.5, the level of methylation increased in the EPI but continued to decrease in the trophectoderm. The level of methylation in the trophectoderm was similar to that described for eutherian post-implantation extraembryonic tissues[22]. Partially methylated domains, a characteristic of the eutherian trophectoderm, were also present in the opossum (Fig. 2b). Thus, relative to that in eutherians, DNA demethylation in opossums occurs late, is transient and modest in the EPI, and is sustained in the trophectoderm. Furthermore, a lower level of methylation and formation of partially methylated domains in the extraembryonic tissue is a conserved feature of therian mammals.

To investigate the mechanisms behind DNA methylation changes during blastocyst development, we examined the mRNA expression of DNA methylation enzymes in our published opossum embryo single-cell RNA sequencing (RNA-seq)[23] data (Fig. 2c) and new multi-omics dataset (Extended Data Fig. 4a), which showed good coherence. Expression of the maintenance methyltransferases *DNMT1A*, *DNMT1B* and the DNMT1 binding partner *UHRF1* was high during cleavage and then declined before blastocyst formation. Notably, between E6.5 and E7.5, the expression of these factors, and of the establishment methyltransferases *DNMT3A* and *DNMT3B*, increased in the EPI concomitant with increased DNA methylation. Conversely, mRNA expression of these enzymes fell in the trophectoderm, concomitant with reduced DNA methylation. The DNMT cofactor *DNMT3L* was not expressed. Expression of the ten-eleven translocation methylcytosine dioxygenase *TET1* was observed; notably, however, levels of expression of *TET2* and *TET3*,

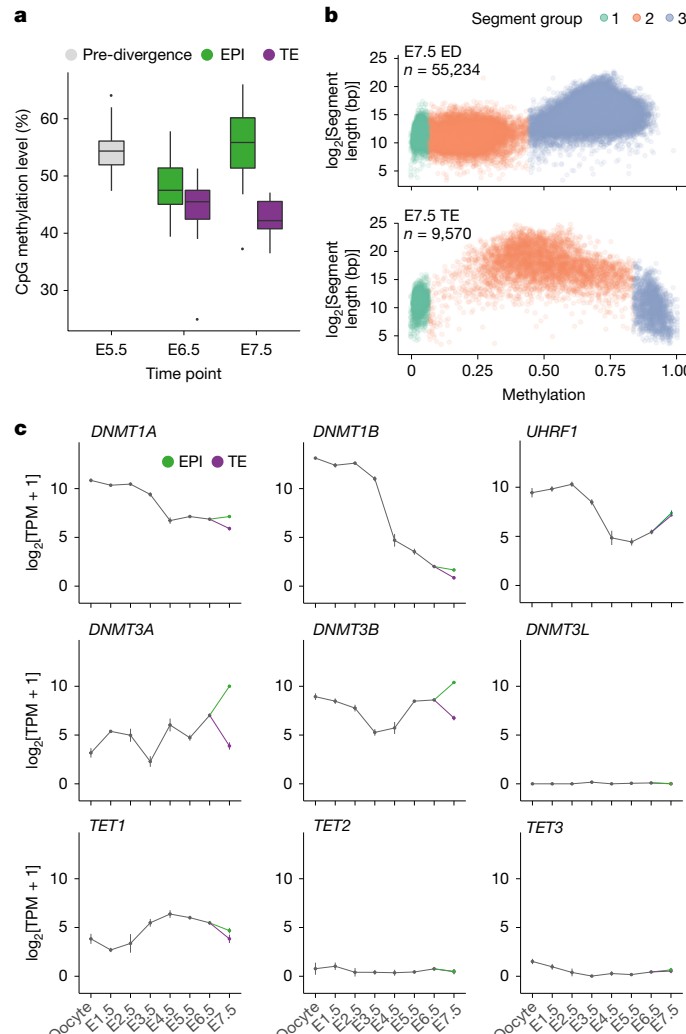

**Fig. 2 | Differential methylation in opossum trophectoderm and EPI. a**, Mean methylation level in blastocyst single cells before and after lineage segregation. $n_{E5.5} = 58$, $n_{E6.5\,EPI} = 19$, $n_{E6.5\,TE} = 19$, $n_{E7.5\,EPI} = 24$, $n_{E7.5\,TE} = 14$. The centre line indicates the median, the bounds indicate the first and third quartiles, and the whiskers indicate 1.5× IQR. **b**, Segmentation of the genome into contiguous regions with similar methylation levels (segments 1–3) according to embryonic disc (ED) and trophectoderm (TE) methylation patterns. Presence of long segments with an intermediate level of methylation in the trophectoderm is indicative of partially methylated domains in this tissue. **c**, Mean expression of DNA methylation machinery derived from previous RNA-seq data[23]. Error bars represent 1.96 × s.e.m. $n_{oocyte} = 6$, $n_{E1.5} = 7$, $n_{E2.5} = 21$, $n_{E3.5} = 62$, $n_{E4.5} = 55$, $n_{E5.5} = 140$, $n_{E6.5} = 201$, $n_{E7.5\,EPI} = 108$, $n_{E7.5\,TE} = 55$.

the latter of which influences methylation in eutherian embryos[18,24–28], were very low. Our findings suggest that differences in *DNMT* enzyme expression could drive the distinct DNA methylation profiles in the EPI and trophectoderm.

## DNA methylation landscape of gametes

We next identified differentially methylated regions (DMRs) in opossum oocytes and sperm. We found 20,800 sperm-specific and 22,921 oocyte-specific DMRs. As in eutherians, oocyte DMRs were relatively enriched in intragenic regions and CGIs, whereas sperm DMRs were enriched in intergenic regions (Fig. 3a). To identify transient and long-term autosomal imprints, we searched for DMRs that exhibited intermediate levels of methylation (40–60%) at E3.5, and tracked their

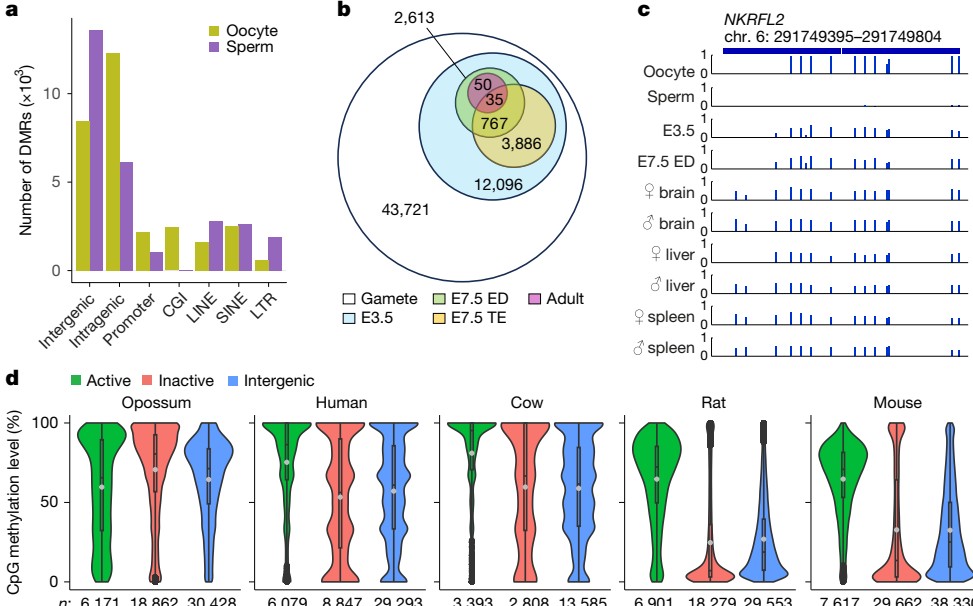

**Fig. 3 | DNA methylation in sperm and oocytes. a**, Genomic distribution of gamete DMRs. **b**, Fate of gamete DMRs during embryogenesis. **c**, DNA methylation levels at individual CpG sites at a representative region of the imprinted *NKRFL2* locus. **d**, DNA methylation profiles in eutherian and opossum oocytes at genes transcribed in oocytes (active), genes not transcribed in oocytes (inactive) and intergenic regions. The centre line indicates the median, the grey circle indicates the mean, the bounds indicate the first and third quartiles, and the whiskers indicate 1.5× IQR.

fate in E7.5 embryonic disc and trophectoderm, and adult brain, liver and spleen. We found 12,096 intermediately methylated regions at E3.5 (Fig. 3b). Around half of these were retained at E7.5, some in a trophectoderm- or embryonic disc-specific manner, and 85 persisted in all 3 adult tissues. As in eutherians, most retained DMRs were of maternal origin (Supplementary Table 2). In adult tissues, we identified 78 DMR-proximal genes, 3 of which are known to be imprinted[29] (*NKRFL2*, *ZFP68* and *RWDD2A*; example locus in Fig. 3c), and the remainder of which are imprinting candidates (Supplementary Table 2).

Eutherian oocytes exhibit divergent DNA methylation patterns, with methylation restricted to the bodies of expressed genes in rodents and extending into non-transcribed regions in other species[3–5,30]. To understand the relationship between oocyte transcription and methylation in opossums, we integrated our methylation datasets with previously published RNA-seq data, and compared our findings with those in other eutherians[23,31–36] (Fig. 3d and Supplementary Table 3). Transcribed genes were heavily methylated across the gene body in eutherians and opossum. Non-transcribed regions (inactive genes and intergenic regions) were methylated at low levels in mice and rats, moderately methylated in humans, and more highly methylated in cows, consistent with previous findings. Opossums exhibited an extreme scenario, with non-transcribed regions exhibiting a very high level of methylation. *DNMT3L* was not expressed in oocytes, indicating that, as in humans[37], de novo DNA methylation in opossum oocytes is *DNMT3L* independent. We also found that non-CpG methylation is relatively enriched in opossum oocytes, as in eutherian oocytes, indicating that this feature is conserved in therians (Extended Data Fig. 4b).

### Inactive X hypomethylation

The active X and inactive X (Xi) exhibit distinct DNA methylation patterns in eutherians. At a chromosome-wide level, the Xi is slightly less methylated than its active counterpart, but the Xi exhibits hypermethylation at CGI promoters[38–42]. We confirmed these findings in adult mouse somatic tissues using a model exhibiting skewed XCI (Extended Data Fig. 5a,b). By contrast, in marsupials, the Xi is hypomethylated at CGIs and transcription start sites[43–48]. However, patterns of DNA

methylation on the X chromosome have not been examined in preimplantation embryos.

We found that in adult opossum tissues, the level of X-chromosome DNA methylation in female animals was approximately half of that in male animals, in agreement with findings in another marsupial, the koala[48] (Fig. 4a and Extended Data Fig. 5c). Allele-specific analysis shows that this was due to hypomethylation of the silenced paternal X chromosome (Extended Data Fig. 5d). Hypomethylation was observed at all genomic features on the Xi, except at XCI escapees[45], where methylation levels were equivalent to those on the active allele (Extended Data Fig. 5e,f). To examine when Xi hypomethylation is established, we assessed X-chromosome DNA methylation levels in embryos (Fig. 4b), which we sexed by quantification of X- and Y-mapping reads (Extended Data Fig. 5g). In the sperm and oocyte, X-chromosome DNA methylation levels were equivalent, demonstrating that the Xi is not inherited from the sperm in a hypomethylated state. Loss of methylation on the Xi occurred gradually during the cleavage stages, with hypomethylation clear from the blastocyst stage (Fig. 4b). Methylation levels therefore differ between chromosomes, being retained on the active X and the autosomes, but lost on the Xi during embryo development.

### A candidate mechanism for imprinted XCI

In most eutherians, XCI in the early embryo is random[49,50]. The mouse is an important exception, with random XCI preceded by silencing of the paternal X imprinted in the preimplantation embryo[51,52]. Imprinted XCI in mice is independent of DNA methylation[53], relying instead on H3K27me3-mediated silencing of the maternal *Xist* allele[54]. The contribution of DNA methylation to XCI imprinting in marsupials is unclear. Methylation of the maternal *RSX* promoter has been observed in fetal opossum tissue[45], but whether this epigenetic mark is inherited from the gamete is unknown. We identified a DMR encompassing the *RSX* promoter that was highly methylated in the oocyte and hypomethylated in sperm (Fig. 4c). Intermediate methylation of this region in female embryos and adult tissues suggests that this pattern is retained, consistent with an instructive role in imprinted XCI.

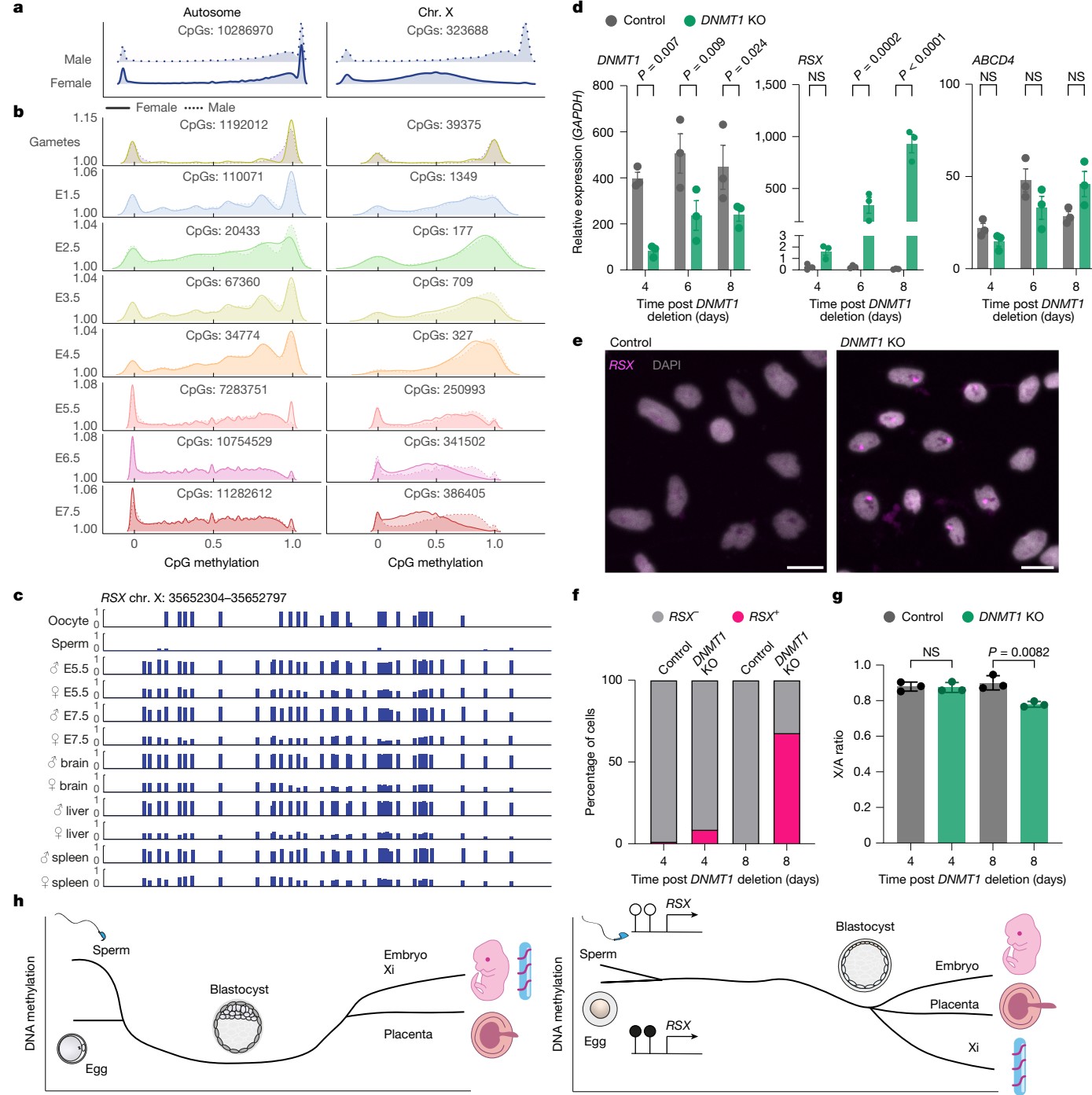

**Fig. 4 | DNA methylation status of the opossum X chromosome and *RSX* locus. a**, Methylation distribution of the autosomes and X chromosome in adult male and female brain represented as density plots showing the distribution of the data and the probability of a variable being a certain value. **b**, Methylation distribution of the autosomes and X chromosome in male and female gametes and embryos. **c**, Methylation at the *RSX* locus in gametes, embryos and adult tissues. **d**, Quantitative PCR analysis in *DNMT1*-knockout (KO) immortalized male fibroblasts. $n_{control} = 3$, $n_{DNMT1\,KO} = 3$. Unpaired *t*-test. Each point represents

the mean of the replicates. Error bars represent s.e.m. **e**, *RSX* RNA fluorescence in situ hybridization in wild-type and *DMNT1*-KO day-8 immortalized male fibroblasts. DAPI, 4′,6-diamidino-2-phenylindole. Scale bars, 20 μm. **f**, Quantification of *RSX* clouds in RNA fluorescence in situ hybridization at 4 and 8 days after *DNMT1* deletion. **g**, X/A gene expression ratios at 4 and 8 days after *DNMT1* deletion. $n_{control} = 3$, $n_{DNMT1\,KO} = 3$. Unpaired *t*-test. Error bars represent s.d. Each point represents the median X/A ratio per replicate. **h**, Schematic of DNA methylation status in eutherian (left) and opossum (right) embryos.

To assess its role in *RSX* regulation, we ablated DNA methylation in bulk immortalized male opossum fibroblasts using CRISPR-mediated deletion of *DNMT1A* and *DNMT1B* (hereafter *DNMT1*). Loss of *DNMT1* reduced methylation genome wide, including at the *RSX* promoter (Extended Data Fig. 6a,b). It also caused ectopic expression of the normally silent *RSX* allele in male cells, as assayed by quantitative PCR

(Fig. 4d). This ectopic expression was accompanied by formation of *RSX* clouds in male cells (Fig. 4e,f).

Ectopic expression of *RSX* in male fibroblasts was associated with suppression of X-gene activity, with X genes over-represented among all downregulated genes (Extended Data Fig. 6c) and the X-to-autosome (X/A) ratio mildly but significantly decreased (Fig. 4g). Loss of *DNMT1*

also led to a decrease in the level of DNA methylation and an increased level of expression of multiple transposable element families (Extended Data Fig. 6d). We also examined expression of the autosomal *H19* locus, which is imprinted in marsupials[55] (*H19* is not present in the opossum assembly and was therefore not recovered in our earlier imprinted screen). The level of *H19* expression increased following *DNMT1* deletion (Extended Data Fig. 6e), suggesting that DNA methylation may be a general mechanism regulating imprinting in opossums.

## Discussion

Here we define the DNA methylation landscape of early development in a marsupial. We demonstrate that opossums do not undergo a eutherian-like global demethylation phase during cleavage (Fig. 4f). DNA demethylation is mild and transient in the EPI but sustained in the trophectoderm. On the basis of our findings, we suggest that in therian mammals, DNA demethylation may play an especially important role in trophectoderm formation. Differences in DNA methylation between the EPI and trophectoderm are presumably achieved through cell-autonomous mechanisms, because the marsupial blastocyst is unilaminar. We propose that this is regulated by differential expression of the DNA methylation machinery. DNA demethylation is also proposed to erase germline-acquired epimutations[7]. This model would predict that epimutations are more stably maintained in marsupials than in eutherians.

We also identify new candidate marsupial imprinted genes that will be useful for investigating conservation and evolution of imprints in therians. In eutherians, imprints are maintained in embryos through the Krüppel-associated box zinc-finger proteins ZFP57 and ZFP445. Of the two, only ZFP445 is conserved in marsupials[56] and may contribute to imprint maintenance in these mammals. We offer a mechanism for marsupial imprinted XCI, mediated by differential methylation of *RSX*. Once methods to epigenetically edit the marsupial genome become available, it would be possible to further test this mechanism by targeted ablation of DNA methylation at the *RSX* promoter. This DNA methylation-mediated mechanism of imprinted XCI differs to that in mice, which achieve imprinted XCI using H3K27me3. We previously identified *XSR*, an *RSX* antisense RNA expressed in oocytes and from the maternal X chromosome in embryos[23]. Although its contribution to imprinted XCI is not established, *XSR* may help establish *RSX* promoter methylation through a mechanism reminiscent of *Tsix*-mediated *Xist* promoter methylation[23].

Our work highlights the strength of the marsupial model to understand evolutionary epigenetics. Given that post-fertilization global DNA methylation erasure is absent in the opossum, and zebrafish[15,16], we propose that this process evolved after the marsupial–eutherian split. In this context, it is noteworthy that *Dppa3* (also known as *Stella* or *Pgc7*) evolved after the marsupial–eutherian divergence[57]. In eutherians, *Dppa3* is important for preimplantation development[58–60], and has been implicated in both passive demethylation and protection against demethylation[57,61–68]. Our current findings are consistent with the hypothesis that evolutionary acquisition of *Dppa3* accompanied the development of oocyte and embryonic DNA demethylation[9,57].

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

## Methods

### Animals

Opossums and mice (C57BL/6J and *Mus spretus*) were maintained in the Francis Crick Institute Biological Research Facility in accordance with UK Animal Scientific Procedures Act 1986 regulations (project licence P8ECF28D9) and subject to Francis Crick Institute's internal ethical review. Additional opossums were housed at the University of Texas Rio Grande Valley under Institutional Animal Care and Use Committee protocol AUP-19-31. No randomization or blinding was performed and no statistical methods were used to predetermine sample size. Mice were housed in individually ventilated cages (GM500, Tecniplast) with a 12:12-h light/dark cycle, a temperature of 20–24 °C and humidity of 55% ± 10%. Adults (8 weeks–6 months) were housed in groups of 3–4 animals, with the different sexes housed separately. Mice had free access to water and food and were provided enrichment activities including rodent balls and nesting boxes. Matings for timed collection of embryos were conducted by placing female mice into the cage of male mice at approximately 17:00. Observation of a vaginal plug the following morning was taken to indicate that mating had occurred.

Opossums were individually housed in Double Decker cages (GR1800, Tecniplast), with male and female animals housed in separate rooms except during mating periods. The temperature of the housing was maintained between 24 °C and 28 °C, and humidity was maintained between 55% and 75%, with a 14 h/10 h light/dark cycle. Opossums had free access to dried food and water, supplemented every second day with live mealworms, and weekly by fresh fruit. To induce oestrous before mating, adult male and female (≥6 months) mice were placed in single-storey rat cages immediately adjacent to each other for 2 days, and then swapped into each other's cages for an additional 2 days. Subsequently, pairs were placed into the same cage and kept together for 10 days, during which period animals were monitored by infrared CCTV camera for mating behaviour[69].

### Collection of gamete, embryo and tissue samples

Mouse E0.5 (11 h post coitum (hpc)) and E3.5 (82–84 hpc) embryos were recovered by flushing the uteri with PBS (Gibco number 14190-094) from a blunt-ended needle under a Leica MC80 dissecting microscope, aspirated using a Stripper pipette with a 275-µm tip (MXL3-STR and MXL3-275, Cooper Surgical), and washed three times through drops of clean PBS. For immunofluorescence, zygotes were fixed in 4% PFA in PBS with 0.2% Triton X-100 for 60 min at room temperature and washed three times in 0.2% Triton X-100 in PBS with 0.01% polyvinyl alcohol (PVA; number P8136, Sigma). For sequencing of blastocysts, the zona pellucida was removed by incubation in acid Tyrode's solution (T1788, Sigma-Aldrich), and then the embryo was washed three times in PBS, snap-frozen on dry ice and stored at −80 °C until processing. Picking-buffer-only negative controls were collected in parallel, and processed through the library preparation procedure to verify the absence of contamination.

Opossum embryos were recovered from the uteri in PBS under a Leica MC80 dissecting microscope at E0.5 (29 hpc), E1.5 (36 hpc), E2.5 (60 hpc), E3.5 (84 hpc), E4.5 (108 hpc), E5.5 (132 hpc), E6.5 (156 hpc) and E7.5 (180 hpc). In the case of oocyte collection, the female animal was mated to a vasectomized male animal, and oocytes were recovered from the uterus at 36 hpc. Embryos were aspirated using a Stripper pipette with a 600-µm tip (MXL3-600, Cooper Surgical), or a micropipette fitted with a 200-µl tip for large E7.5 embryos, and washed three times in PBS. Samples were imaged using a Leica DMIL LED microscope with a 20× objective and then further processed for sequencing or immunofluorescence. The shell coat was perforated with 1-µm dissecting needles (10130-20, FST) and incubated in 5 mg ml⁻¹ protease in PBS (P8811-100MG, Sigma) at 32 °C for between 2 and 7 min. The sample was transferred into fresh PBS, and any remaining mucoid coat was removed by disaggregation with dissecting needles.

For sequencing, the oocytes and E0.5–7.5 embryos were washed through three drops of fresh PBS, dispensed to 0.2 µl tubes, and snap-frozen on dry ice and stored at −80 °C until further processing. We collected multiple single embryos for each time point, and in addition collected pooled litters of oocyte, E2.5 and E3.5 samples (Supplementary Table 1). For E7.5 embryonic disc and trophectoderm samples, after removal of the shell coat, the embryonic disc was isolated from the trophectoderm T by dissection with 1-µm needles, washed through three drops of PBS, and snap-frozen as above. For single-cell sequencing, embryos were washed through three drops of fresh PBS with BSA, and incubated in TrypLE (12604013, ThermoFisher) diluted to 0.5× in PBS for 2–8 min at 35 °C. Following TrypLE incubation, embryos were moved to PBS with BSA and disaggregated to single cells by repeated pipetting through a narrow-bore pipette, before nucleosome, methylome and transcriptome (NMT)-sequencing processing. Picking-buffer controls were included as described above. For immunofluorescence, E0.5 zygotes were fixed and washed as above for mouse embryos.

For collection of sperm from mouse and opossum adult male animals, epididymides were dissected from the testes and rinsed in PBS. Cauda epididymides were transferred to 2 ml Bigger–Whitten–Whittingham buffer (opossums) or TYH buffer (mice). Several small incisions were made, followed by incubation at 37 °C for 30 min to facilitate sperm swim out. The swim-out was diluted approximately 1:10 in PBS, and visualized on a dissecting microscope using a dark field. Individual sperm were aspirated using a Stripper pipette with a 100-µl tip (MXL3-100, Cooper Surgical) and washed three times in PBS before collection in 5 µl RLT Plus (1053393, Qiagen) containing 2% β-mercaptoethanol. For collection of pools of sperm, multiple sperm cells were picked in one pipette and processed together through washing, collection and freezing as for single sperm cells. Picking-buffer controls were included as described above.

For collection of genomic DNA from adult mice, and genomic DNA and RNA samples from adult opossums, brain, liver and spleen tissues were dissected into ≈10-mm pieces, snap-frozen in liquid nitrogen, and stored at −80 °C. To facilitate allele-specific analyses, a cross was set up with parent animals from the LL2 opossum stock, and tissue samples were collected from the parents for whole-genome sequencing (WGS) and genomic variant identification, and from three male and three female littermates for BS-seq and RNA-seq. Mouse tissue samples were collected from a model exhibiting skewed XCI through a targeted *Xist* deletion[70] (that is, C57BL/6J *Xist^tm1Jae^* × *Mus spretus* F₁ hybrid cross). We collected three heterozygous female offspring in which XCI is completely skewed, and three male littermates with *Xist* deleted. Frozen tissue was pulverized using a pestle and mortar pre-cooled on dry ice, and used for genomic DNA extraction with the PureLink Genomic DNA Mini kit (K18290-02, Invitrogen) or RNA extraction using the Ambion RNAqueous-Micro Kit (AM1931, ThermoFisher). Samples were then processed for WGS, BS-seq or RNA-seq library preparation.

### BS-seq

Library preparation was performed with oligonucleotides compatible with the NEBNext library preparation kit (E7535S) following an established method[71] and described briefly here. For gamete and embryo samples, samples were lysed in 2.5 µl RLT Plus and processed following the low-input library method[71]. For brain, liver, spleen and fibroblast samples, 6 ng of genomic DNA was used to prepare libraries following the bulk method[71]. Samples were spiked with 6 fg of Lambda DNA (D152A, Promega) and bisulfite-converted using the EZ Methylation Kit (D5020, Zymo). Bisulfite-converted DNA was purified using the PureLink PCR Purification Column Kit (K310050, Invitrogen) with an additional treatment with M-desulfonation buffer (EZ Methylation Kit, Zymo). Samples were eluted into a mixture containing 0.4 µM Preamp primer (5′-CTACACGACGCTCTTCCGATCTNNNNNN-3′) and submitted to one (brain, liver and spleen samples) or five (gamete and embryo samples) rounds of pre-amplification with Klenow exo− (M0212M,

NEB). Unused oligonucleotides were degraded by incubation with exonuclease I (M0293L, NEB), and samples were purified with Ampure XP beads (A63881, Beckman Coulter). Samples were resuspended in a mixture containing 0.4 µM Adaptor 2 Oligo for NEB indices (5′-CAGACGTGTGCTCTTCCGATCTNNNNNN-3′) and adaptor-tagged by incubation with Klenow exo−. Samples were purified using Ampure XP beads and resuspended in a mixture containing 0.2 µM NEBNext Universal Adaptor and 0.2 µM NEBNext Index Adaptor (E7535S, NEB), and library amplification was performed using KAPI Hifi HotStart polymerase (KK2502, KAPA Biosystems). Varying numbers of PCR cycles were performed depending on input amount (opossum oocytes and E1.5–E5.5 embryos: 19 cycles; mouse E3.5 embryos and opossum E6.5 and E7.5 embryos: 10–14 cycles; sperm: 10–18 cycles; brain, liver and spleen: 10 cycles). Library sequencing was carried out by the Francis Crick Institute Advanced Sequencing Facility (ASF). Gamete and embryo libraries were sequenced (100-base-pair (bp) paired end) on an Illumina HiSeq 4000, yielding between 47 million and 337 million reads per library. Brain, liver and spleen libraries were sequenced (150-bp paired end) on an Illumina HiSeq 4000, yielding between 198 million and 363 million reads per library.

## RNA-seq

Purified RNA was submitted to the Francis Crick Institute ASF for preparation of cDNA using the SMART-Seq v4 Ultra Low Input RNA Kit (634894, Takara), followed by library preparation using the Nextera XT DNA Library Preparation Kit (FC-131-1096, Illumina). Sequencing (100-bp paired end) was performed on an Illumina HiSeq 4000, generating between 54 million and 156 million reads per library.

## NMT-seq

Disaggregated single cells were deposited into individual wells of a 96-well plate, and NMT-seq libraries were prepared as previously described[71]. In brief, cells were incubated with M.CviPI for 15 min, followed by separation of mRNA and genomic DNA using Oligo-dT beads, and subsequent preparation of RNA-seq and BS-seq libraries. Sequencing (100-bp paired end) was performed on an Illumina HiSeq 4000.

## DNMT1-knockout opossum fibroblasts

Primary opossum fibroblasts were derived from a newborn male animal and immortalized using SV40-tag virus infection. Single-cell clonal selection was performed to identify an euploid cell line. Opossum fibroblasts were maintained in DMEM (Gibco) supplemented with 20% fetal bovine serum, 1% GlutaMax (Gibco), 1% sodium pyruvate (Gibco) and 1% penicillin–streptomycin (Gibco, 10,000 U ml⁻¹) and were routinely tested and found to be negative for mycoplasma. Single-guide RNAs (sgRNAs) targeting all opossum *DNMT1* paralogues (*DNMT1A*, *DNMT1B* and *DNMT1Ψ*) were designed using the online tool CRISPRdirect (https://crispr.dbcls.jp/): *DNMT1* gRNA 1: 5′-TCTGAAGGCTTTCATCAAGC-3′; *DNMT1* gRNA 2: 5′-CATTGTGGGCCATTGAAATG-3′. sgRNAs were annealed and ligated into the targeting plasmid px333-puro. The plasmid px333-puro was obtained after cloning a puromycin-resistance cassette isolated from px459v2 (gift from F. Zhang, Addgene number 62988) into the px333 vector (gift from A. Ventura, Addgene number 64073)[72].

Immortalized opossum fibroblasts were seeded onto gelatin-coated wells of 6-well plates. The following day, fibroblasts were transfected using PEI MAX (49553-93-7). A 2 µg quantity of plasmid (with sgRNAs or empty plasmid for negative control) was added to 200 µl of Opti-MEM (Gibco) and 8 µl of PEI MAX (1 mg ml⁻¹). One day after transfection, puromycin (2.5 µg ml⁻¹) was added for 48 h to select successfully transfected cells. Control and *DNMT1*-knockout cells were fixed for RNA fluorescence in situ hybridization (RNA FISH) or frozen down for quantitative PCR with reverse transcription (qRT–PCR) at specific time points.

## RNA FISH

Cells were washed in cold PBS and treated with ice-cold permeabilizing solution (0.5% Triton X-100, 2 mM vanadyl ribonucleoside complex in PBS) for 10 min. After fixation using ice-cold 4% PFA in PBS for 10 min, cells were rinsed in ice-cold PBS twice, dehydrated through ice-cold 70%, 80%, 95% and 100% ethanol for 3 min each, and air-dried. BAC VM-18-303M7 (CHORI) was used for *RSX* RNA FISH. BAC DNA was labelled using Nick Translation Kit (Abbott) with fluorescent nucleotides (spectrum orange-dUTP; 02N33-050, Abott), and cells were hybridized with a denatured mix of probes along with 1 µg salmon sperm DNA in hybridization buffer (50% formamide, 10% dextran sulfate, 1 mg ml⁻¹ PVP, 0.05% Triton X-100, 0.5 mg ml⁻¹ BSA, 1 mM vanadyl ribonucleoside complex in 2× SSC) at 37 °C overnight in a humid chamber. Stringency washes were performed on a hot plate, three times for 5 min in 50% formamide in 1× SSC (pH 7.2–7.4) preheated to 45 °C, and three times for 5 min in 2× SSC (pH 7–7.2) preheated to 45 °C. Cells were mounted in antifade containing DAPI (Vector) with a coverslip and stored at −20 °C.

## RNA purification and qRT–PCR

RNA from control and *DNMT1*-knockout male opossum fibroblasts across three different time points and three replicates was purified using RNAqueous-Micro Total RNA Isolation kit (Invitrogen, AM1931). Purified RNA was retrotranscribed using the Maxima First Strand cDNA synthesis kit (Thermo Scientific, K1641). The following primers were used to assess gene expression in RT–PCR assays using PowerUp SYBR Green (Applied Biosystems, A25780): *RSX* (5′-AGAAGGGACCCCAAGACAC-3′, 5′-TGGGTCACTTCCACTTCCTC-3′); *DNMT1* (5′-GACGCAGTAACACTGGAGCA-3′, 5′-ATCCCATTCCAACCTTCCAT-3′); *H19* (5′-TCCAGCAGCAGTCAGTGAAC-3′, 5′-TCATCCATCCATGAGCAGAG-3′); *ABCD4* (5′-ATCGATAATCCGGACCAGCG-3′, 5′-ATGATCAGCTTGCTGGCCAT-3′), *GAPDH* (5′-TAAATGGGGAGATGCTGGAG-3′, 5′-ATGCCGAAGTTGTCGTGAA-3′).

## Fibroblast bulk RNA-seq

Libraries from control and *DNMT1*-knockout RNA samples from day 4 and 8 were prepared with the NEBNext Ultra II Directional PolyA mRNA kit according to the manufacturer's instructions. Libraries were sequenced on the Illumina NovaSeq 6000 system (paired end, 100-bp read length). Raw RNA-seq reads were processed using the RNA-seq nf-core pipeline (v3.2); star_rsem was used to generate raw reads counts. The read counts were processed in R using the DESeq2 package (v1.36). Genes expressed at very low levels were filtered out by applying a rowSums filter of ≥5 to the raw counts table. Raw counts were normalized using the DESeq() function, specifying ~genotype_day in the design formula. log₂[fold change] and adjusted *P* values between *DNMT1* knockout and control were calculated using the lfcShrink() function in DESeq2, specifying type = 'ashr', analysing day 4 and day 8 separately.

X/A ratios were calculated using the median expression of X and autosomal genes in each sample after filtering out genes expressed at low levels (transcripts per million (TPM) of <1). X/A ratios between control and *DNMT1*-knockout samples were compared by Student's *t*-test. Repetitive element expression was analysed as above for opossum embryos.

## WGS

Samples of 3 µg of genomic DNA prepared from ear snips from one male and one female opossum from the LL2 stock were submitted to the Francis Crick Institute ASF for library construction (KAPA HyperPlus). Libraries were sequenced (150-bp paired end) on a HiSeq 4000, producing 256,834,288 reads (99.23% mapped) for the female animal and 122,371,579 reads (99.27% mapped) for the male animal. Data were used for identification of genomic variants, described below.

## Preparation of a modified MonDom5 reference genome and annotations

We modified the MonDom5 reference genome to include a gap-filling long-read sequence of the *RSX* locus[73]. We prepared modified gene, repeat and CGI annotation files with corrected coordinates on the gap-filled X chromosome. We also included annotation for both opossum *DNMT1* paralogues[17] in our modified gene annotation file.

## Expression analysis of opossum DNA methylation factors and repetitive elements

Opossum embryo RNA-seq data[23] and *DNMT*1-knockout RNA-seq data were mapped to the modified MonDom5 reference genome and ASM229v1 reference genome, respectively, using HISAT2 (ref. 74) with the command hisat2 -3 0 -5 9 --fr --no-mixed --no-discordant, and read summarization at genes was performed using the Rsubread package[75], excluding multi-mapping reads. Fragments per kilobase of transcript per million mapped read (FPKM) values were calculated using the scater package[76]. Read summarization at repetitive elements and calculation of counts per million were performed using Telescope[77]. Line plots showing mean and standard error of gene and repeat expression were generated using ggplot2.

## Analysis of methylation at repetitive elements

To generate a bigwig file per time point, the filtered methylKit csv files were converted from csv to bedGraph using awk and then converted to bigwig using the UCSC bedGraphToBigWig utility. Genomic coordinates of the transposable elements, specifically L1, MIR and ERV1, were obtained from the mondom5 RepeatMasker GTF file, selecting for those in the 'forward' orientation. Using deepTools[78], the computeMatrix function in scale-regions mode was used to calculate the methylation score, using the parameters --binSize 50 --averageTypeBins mean -a 1000 -b 1000. The plotProfile function was used to generate the profile plot for each transposable element, using the parameters --perGroup --yMin 0 --yMax 1.

## Methylation segmentation of E7.5 embryonic disc and trophectoderm

The methSeg function from the R package MethylKit[79] was used to divide the genome into contiguous stretches of similar methylation level in E7.5 embryonic disc and trophectoderm samples. Parameters used were minSeg = 5, G = 1:3, join.neighbours = TRUE. Individual CpG sites with a minimum coverage of five reads in both samples were used for the analysis. The length and average methylation level of each segment were plotted using ggplot2.

## Single-cell methylation analysis

Single-cell RNA-seq data were aligned to the opossum ASM229v1 genome, using the nf-core rnaseq pipeline (3.2), using the parameters –aligner star_rsem –bam-csi-index. Raw counts were loaded into Seurat (4.3.0)[80] for analysis in R (4.2.2). In total, there were 74 E5.5 samples, 142 E6.5 samples and 42 E7.5 samples. Each dataset was normalized using NormalizeData, and then datasets were integrated together using the functions SelectIntegrationFeatures, FindIntegrationAnchors and then IntegrateData. The integrated dataset was scaled used ScaleData. Principal component analysis was applied using RunPCA. Uniform manifold approximation and projection was estimated for the integrated dataset using RunUMAP. Module scores for EPI and trophectoderm were calculated using the AddModuleScore function, with the following genes used as markers for each cell lineage. EPI: *NANOG*, *PRDM14*, *POU5F1* and *POU5F3*; trophectoderm: *GATA2*, *GATA3*, *TEAD4*, *AQP3* and *KLF4*.

Single-cell BS-seq data were trimmed using the TrimGalore![81] command trim_galore --clip_R1 6 --three_prime_clip_r1 6, mapped using Bismark[82] with the command bismark --non_directional --un --ambiguous --multicore 2 and deduplicated using the command deduplicate_bismark. Methylation information was extracted with the command bismark_methylation_extractor --comprehensive --multicore 2 --bedGraph --CX --cytosine_report --nome-seq. Average CpG methylation was calculated per cell and plotted according to the cell lineage determined for the corresponding RNA-seq library (module score, above).

## Analysis of methylation in eutherian oocytes

Raw RNA-seq and BS-seq data were processed using the nf-core rnaseq (3.12) and methylseq (2.5.0) pipelines, respectively. Gene bodies and intergenic regions for each organism were identified using the GenomicFeatures[83] R package. Following a methodology similar to that in ref. 34, genes were classified as active if their TPM was >5 and inactive if their TPM was ≤1, and Bismark CpG methylation calls were imported into R using the methylKit[79] R package, then destranded, pooled by sample condition and filtered for CpG sites with a minimum coverage of three reads. The regionCounts function was used to calculate the methylation level across active genes, inactive genes and intergenic regions. These were visualized as violin plots used ggplot2 (ref. 84).

## Identification of genomic variants

WGS data from LL2 parent opossums were used to identify genomic variants. WGS reads were trimmed using TrimGalore![81] with the command trim_galore --cores 4 --paired --fastqc --gzip --retain_unpaired --clip_R1 10 --clip_R2 10 --three_prime_clip_R1 5 --three_prime_clip_R2 5. Libraries were mapped to the MonDom5 reference genome using BWA-MEM[85] with the command bwa mem -t 32 -M -R. Paired and unpaired mapped reads were then merged, sorted and indexed using SAMTools[86]. Variants were called using the GATK best practices pipeline[87]. For the base recalibration step, known variants were not available for opossums. Therefore, variants were initially called independently with three pipelines: BCFtools[88], Varscan[89] and GATK[87]. Variants identified by all three pipelines were considered high-confidence variants and were used for GATK base recalibration. Subsequently, variants were called for each opossum, and then combined, and genotypes were annotated to produce a variant call file. BEDtools maskfasta was used to create an N-masked version of the MonDom5 reference genome from the complete set of 25 million variants. Using a custom R script, variants were filtered to include only hemizygous and heterozygous single nucleotide polymorphisms (SNPs). The resultant 2 million SNPs were used in the SNPsplit pipeline[90].

For mice, genomic variant data (41,668,158 SNPs) were derived from the C57BL/6J and *M. spretus* genomes (Mouse Genomes Project[91]), and the program SNPsplit[90] was used to generate an mm10 reference genome in which parental genomic variants were N-masked.

## Methylation analysis

BS-seq reads were trimmed using the TrimGalore![81] command trim_galore --clip_R1 6 --three_prime_clip_r1 6. Reads were mapped to the mm10 (mouse) or MonDom5 (opossum) reference genome using Bismark[82] on single-end mode with the command bismark --non_directional --un –ambiguous. For adult libraries, reads were mapped to the N-masked genomes, and bam files were generated for all mapped reads as well as for allele-specific reads using SNPsplit[90] with the command SNPsplit --bisulfite –conflicting. Library statistics (Supplementary Table 1) were extracted from TrimGalore! and Bismark output reports. Bam files were deduplicated and methylation calls were extracted using Bismark. CpG methylation calls were imported into R[92] using the package methylKit[79], and principal component analysis was performed using all CpG methylation calls for each individual sample. Data were destranded and pooled by sample condition, and the number of CpG sites captured at different coverage thresholds was calculated. For subsequent analyses, data were filtered for CpG sites with a minimum coverage of five reads. Methylation distribution histograms, mean methylation plot and genomic coverage plots were generated using

ggplot2 (ref. 84). This workflow was independently reproduced to ensure that the results were accurate and robust.

## Analysis of gamete DMRs

DMRs between gametes were identified as follows: 100-bp non-overlapping tiles covered by a minimum of three reads in both oocyte and sperm samples and containing a minimum of one CpG in a tile were identified. The methylation level of each tile was compared between oocyte and sperm using the diffMeth function from the R package methylKit[79], with the parameters difference = 80 and qvalue = 0.01. Gamete DMRs with putative transient or life-long retention of differential methylation were defined as tiles with intermediate methylation levels (40–60% methylated), in either E3.5 and E7.5 embryonic disc or trophectoderm (transient embryonic or trophectoderm DMRs), or in E3.5, E7.5 embryonic disc, and all three of brain, liver and spleen (life-long DMRs). The gene nearest to each life-long DMR was identified using the nearest function from the R package GenomicRanges[83], and this list was manually checked for previously reported marsupial imprinted genes[29,93]. To retain maximum read coverage in low-input embryo samples, all libraries per time point were in silico-pooled for this analysis. The dataset therefore included male and female embryos, precluding analysis of sex differences or the X chromosome.

## Sex-specific methylation analysis

CpG methylation calls were destranded, pooled by sample condition (sex and tissue) and filtered for minimum coverage of five reads in all conditions. For embryo samples, sex was inferred from the ratio of reads mapping to chromosome X and pseudo-Y (coding sequence of 19 known opossum Y-chromosome genes)[94,95]. Autosomes and X-chromosome allelic methylation distributions were represented as ridgeline plots using the R package ggridges[96].

## Allele-specific methylation analysis

Allele-specific CpG methylation calls were destranded, pooled by sample condition (sex, tissue and parental genome) and filtered for minimum coverage of five reads in all conditions. To avoid loss of X-linked sites in female samples due to low coverage of the X chromosome in male files, data import and filtering for X chromosomes was performed separately with the coverage parameters as for autosomes, but excluding paternal genome files from male samples. Ridgeline plots were generated using ggridges[96] as above.

## Escape-gene methylation analysis

To generate a list of genes expressed in each tissue, RNA-seq reads were trimmed using TrimGalore![81] with the command trim_galore --paired -- clip_R1 10 --clip_R2 10. Trimmed reads were aligned to the N-masked MonDom5 reference genome using HISAT2 (ref. 8) with the command hisat2 --no-softclip --no-mixed --no-discordant. Mapped files were converted to bam files and merged by sample using SAMtools[86]. Reads overlapping annotated genes were quantified using the command featureCounts from the R package Rsubread[75], excluding multi-mapping reads. Files were merged by biological replicate, and FPKM values were calculated using the R package DESeq2 (ref. 97) using the robust median ratio method. Gene models annotated as pseudogenes were excluded, and a threshold of FPKM > 1 was imposed. Expressed genes were categorized as escaping or subject to XCI on the basis of published work[45]. Methylation level at genes was calculated using the methylKit functions regionCounts and percMethylation, and represented as violin plots for each category using ggplot2 (ref. 84).

## Immunofluorescent staining of mouse and opossum embryos

Embryos were permeabilized in 0.5% Triton X-100 in PBS at 4 °C overnight, followed by three washes in 0.2% Triton X-100 in PBS with 0.1% PVA (0.2% TX–PBS–PVA). After 3.5 M HCl treatment for 30 min at room temperature, embryos were washed three times in 0.2% TX–PBS–PVA,

blocked for 1–4 h at room temperature in 3% BSA in 0.2% TX–PBS–PVA (0.22-μM-filtered, blocking solution) and incubated overnight at 4 °C in primary antibody in blocking solution. Antibodies used were 5mC (BI-MECY-0100, Eurogentec) at 1:100, 5hmC (75-268, NeuroMab) at 1:1,000, H3K9me3 (07-442, MerckMillipore) at 1:200, H3 (ab1791, Abcam) at 1:100. Embryos were washed three times in 0.2% TX–PBS–PVA and incubated in Alexa Fluor-conjugated secondary antibodies at 1:250 in blocking solution for 2 h at room temperature. Three further 0.2% TX–PBS–PVA washes were performed, and DNA was counterstained with 10 μg ml$^{-1}$ propidium iodide (P4170, Sigma) for 1–2 h at room temperature. Samples were washed three times with PBS with 0.1% PVA and mounted in Vectashield (H-1000-10, Vector Laboratories). Samples were imaged using an LSM710 confocal microscope with 1-μm z-sections. Images were processed and fluorescence intensity was measured using Fiji[98].

## Reporting summary

Further information on research design is available in the Nature Portfolio Reporting Summary linked to this article.

## Data availability

BS-seq and RNA-seq data have been deposited at the Gene Expression Omnibus (GEO) under the accession number GSE206499. WGS data have been deposited at the Sequence Read Archive under the accession number PRJNA819000. Publicly available datasets analysed in this study are accessible via the Gene Expression Omnibus under the accession numbers GSE163620, GSE71434, GSE101571, GSE163620 and GSE71985; DDBJ under the accession numbers DRA006642 and DRA000570; and ArrayExpress under the accession number E-MTAB-7515. Source data are provided with this paper.

## Code availability

Code used in preparation and analysis of data and the generation of figures is available via GitHub at https://github.com/bleeke/opossum-methylation.

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

**Acknowledgements** We thank the Francis Crick Institute Biological Research Facility, Advanced Sequencing Facility, Bioinformatics and Biostatistics Facility and High-Performance Computing team, M. Sangrithi, N. Fogarty, S. Clark, W. Reik, S. Smallwood, G. Kelsey, P. Skoglund and K. Niakan for advice on the project, and D. Odom and members of the laboratory of J.M.A.T. for comments and discussion on the manuscript. Work in the laboratory of J.M.A.T. is supported by the European Research Council (CoG 647971) and the Francis Crick Institute, which receives its core funding from Cancer Research UK (FC001193), the UK Medical Research Council (FC001193) and the Wellcome Trust (FC001193). Some of the work at the University of Texas Rio Grande Valley was conducted in facilities constructed with support from the NIH grant C06 RR020547. Research in the laboratory of R.J.O. is supported by grants from the Bernice Bibby Research Trust and The Paradifference Foundation, the Wellcome Trust through the research equipment grant (212917/Z/18/Z), the NIHR Biomedical Research Centre at Guy's and St Thomas' Foundation Trust, the MRC (MR/V038664/1) and the King's College London Innovation Fund. A.T. is supported by funding from the European Research Council under the European Union's Horizon 2020 research and innovation programme (grant agreement number 101098236 ERC-2022-AdG). For the purpose of open access, the author has applied a CC BY public copyright licence to any author accepted manuscript version arising from this submission.

**Author contributions** B.J.L. and W.V. are equal first authors, and S.O. and J.Z. are equal second authors. B.J.L. and J.M.A.T. conceived and designed the project and wrote the manuscript. B.J.L. performed gamete, embryo, single-cell and adult tissue collections, BS-seq, RNA-seq, WGS and computational analyses. W.V. performed computational analyses. J.Z. performed adult tissue collections, WGS and computational analyses. S.O. performed embryo immunostaining and fibroblast RNA FISH. S.O. generated *DNMT1*-knockout fibroblasts. S.M. performed single-cell collections, *DNMT1*-knockout experiments and analyses. A.C. imaged and quantified RNA FISH experiments. S.K.M. performed gamete collections. D.M.S. processed NMT-seq libraries. F.D. made initial findings of the *RSX* DMR. J.L.V. provided additional opossum material. O.O. managed the animal colonies. A.T. performed computational analyses. J.M.A.T., J.L.V. and R.J.O. acquired funding.

**Funding** Open Access funding provided by The Francis Crick Institute.

**Competing interests** The authors declare no competing interests.

**Additional information**
**Correspondence and requests for materials** should be addressed to Bryony J. Leeke or James M. A. Turner.

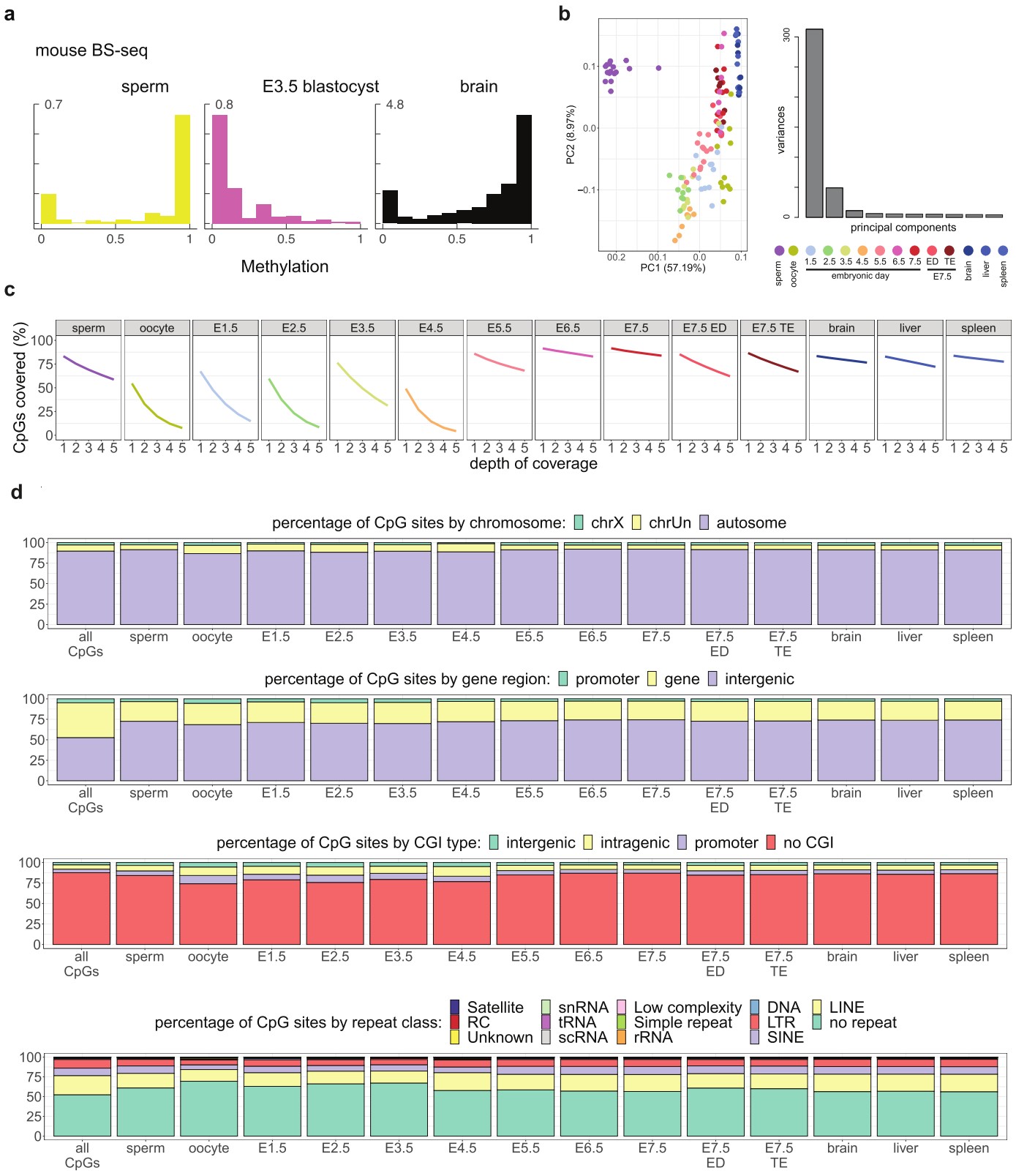

**Extended Data Fig. 1 | Description of mouse and opossum BS-seq data.**
**a**. Histograms of methylation distribution at CpG sites captured at ≥ 5x coverage in mouse sperm (n = 2 libraries of ~100 sperm, in silico pooled), E3.5 embryos (n = 3, in silico pooled), and adult brain (n = 2, in silico pooled) (please note variable scales on y-axes). **b**. Principal component analysis of opossum methylation in all samples. **c**. Percentage of CpG sites captured at different coverage thresholds. **d**. Percentage of captured CpG sites for different chromosomes and genomic features.

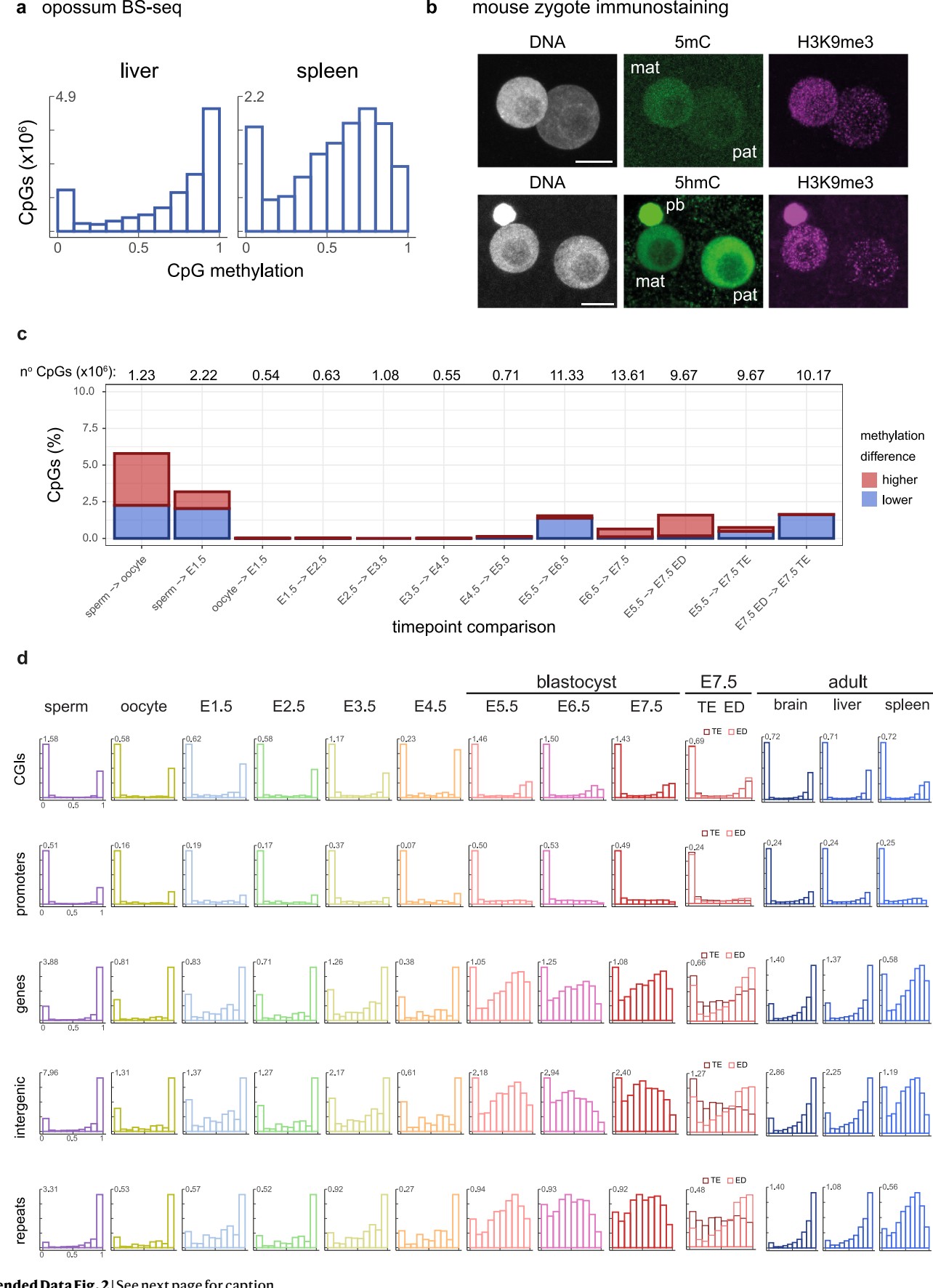

**Extended Data Fig. 2** | See next page for caption.

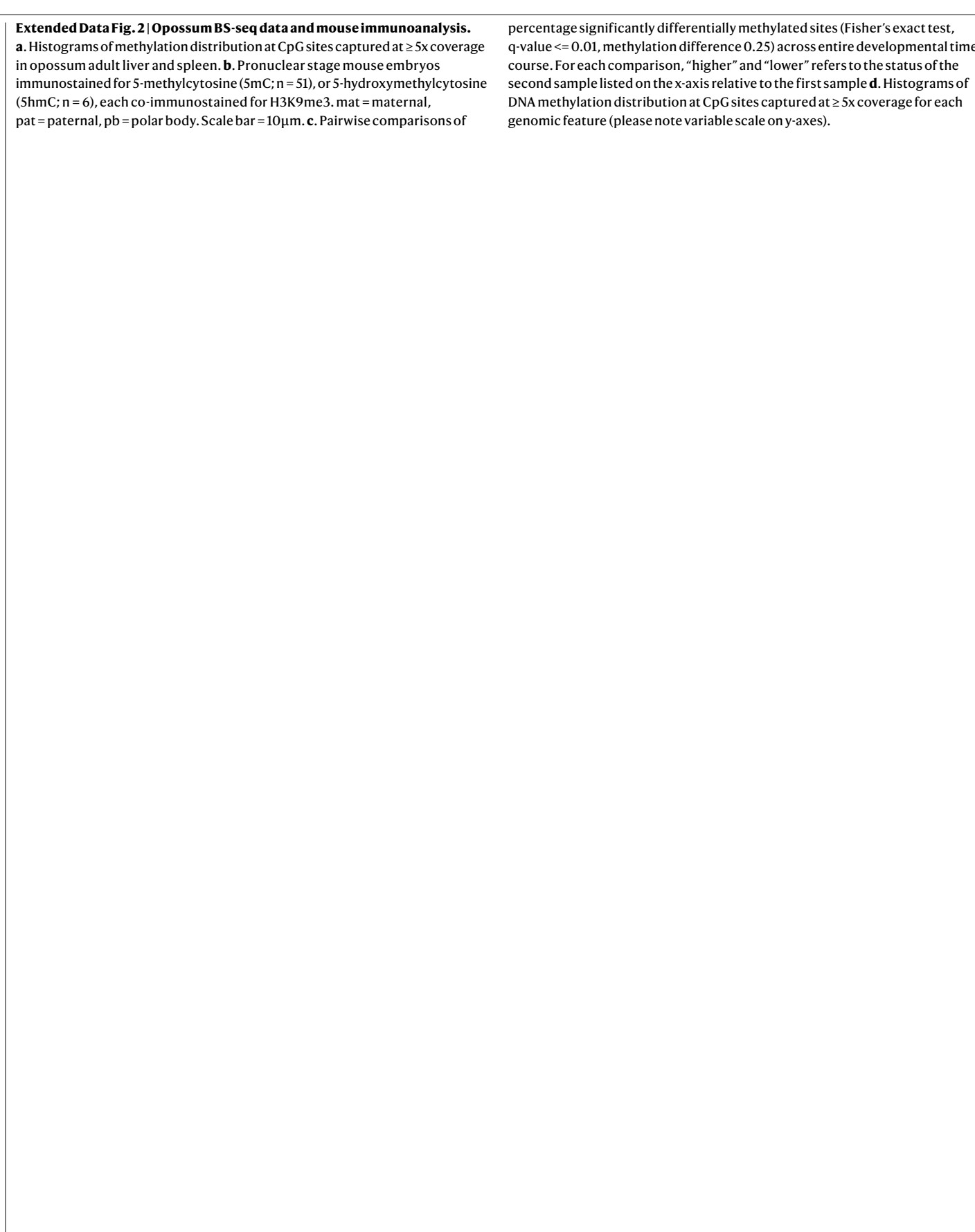

**Extended Data Fig. 2 | Opossum BS-seq data and mouse immunoanalysis.**
**a**. Histograms of methylation distribution at CpG sites captured at ≥ 5x coverage in opossum adult liver and spleen. **b**. Pronuclear stage mouse embryos immunostained for 5-methylcytosine (5mC; n = 51), or 5-hydroxymethylcytosine (5hmC; n = 6), each co-immunostained for H3K9me3. mat = maternal, pat = paternal, pb = polar body. Scale bar = 10μm. **c**. Pairwise comparisons of percentage significantly differentially methylated sites (Fisher's exact test, q-value <= 0.01, methylation difference 0.25) across entire developmental time course. For each comparison, "higher" and "lower" refers to the status of the second sample listed on the x-axis relative to the first sample **d**. Histograms of DNA methylation distribution at CpG sites captured at ≥ 5x coverage for each genomic feature (please note variable scale on y-axes).

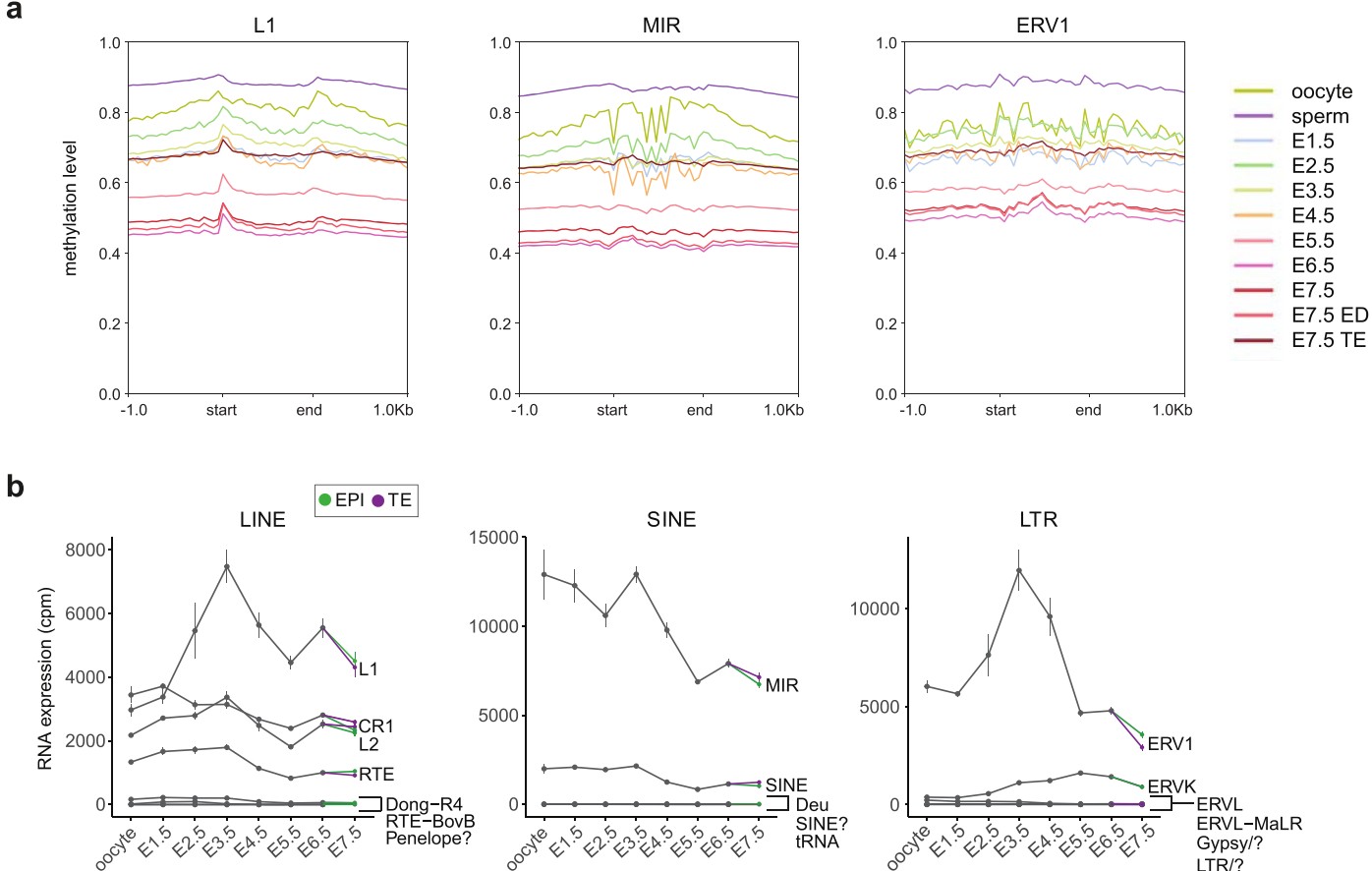

**Extended Data Fig. 3 | DNA methylation and expression of repetitive elements in opossum embryos. a**. Metaplots of methylation across L1, MIR, and ERV1 loci in opossum embryos. **b**. Expression of LINE, SINE and LTR repetitive elements in opossum embryos. Repetitive element subfamilies are labelled according to Repeatmasker nomenclature, including '?' notation for presumptive subfamily. $N_{oocyte} = 12$, $N_{E1.5} = 14$, $N_{E2.5} = 42$, $N_{E3.5} = 124$, $N_{E4.5} = 110$, $N_{E5.5} = 280$, $N_{E6.5} = 402$, $N_{E7.5 EPI} = 216$, $N_{E7.5 TE} = 110$. Error bars = 1.96*SE. Each point represents the mean of the replicates.

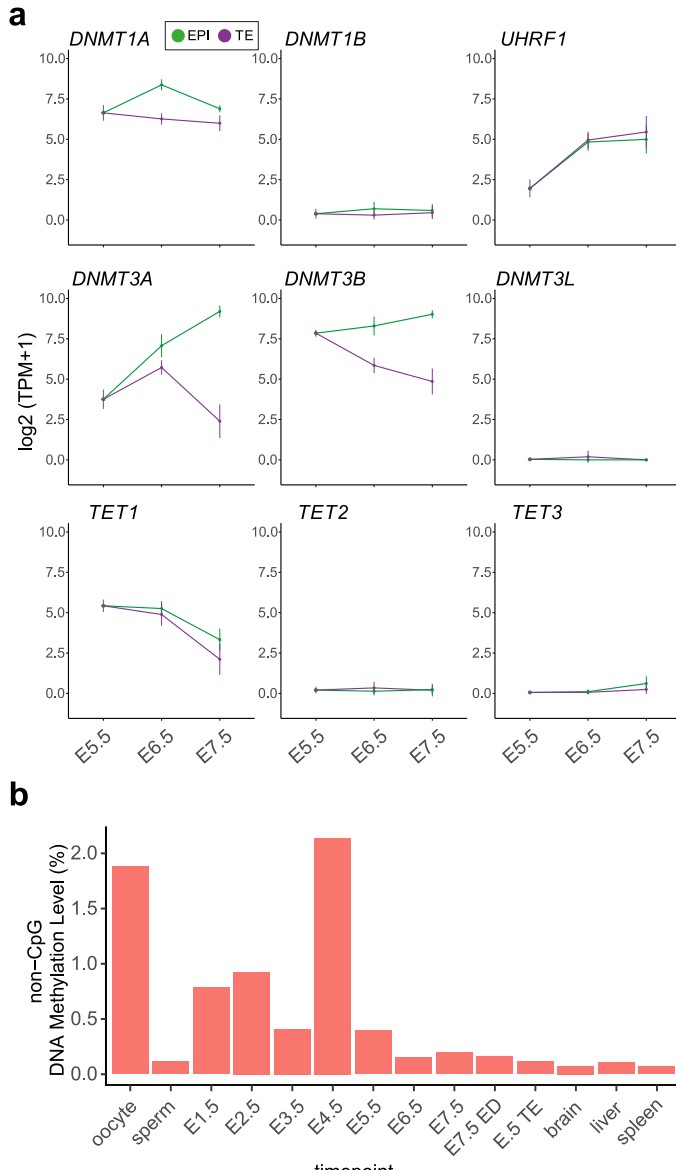

**a**

**b**

**Extended Data Fig. 4 | Expression of methylation enzymes and non-CpG methylation levels. A**. Expression of methylation enzymes in EPI and TE using the multi-omics dataset. $N_{E5.5} = 58$, $N_{E6.5\,EPI} = 19$, $N_{E6.5\,TE} = 19$, $N_{E7.5\,EPI} = 24$, $N_{E7.5\,TE} = 14$. Error bars = 1.96*SE. Each point represents the mean of the replicates. **b**. Levels of non-CpG methylation (CHH and CHG sites) in opossum gametes and embryos.

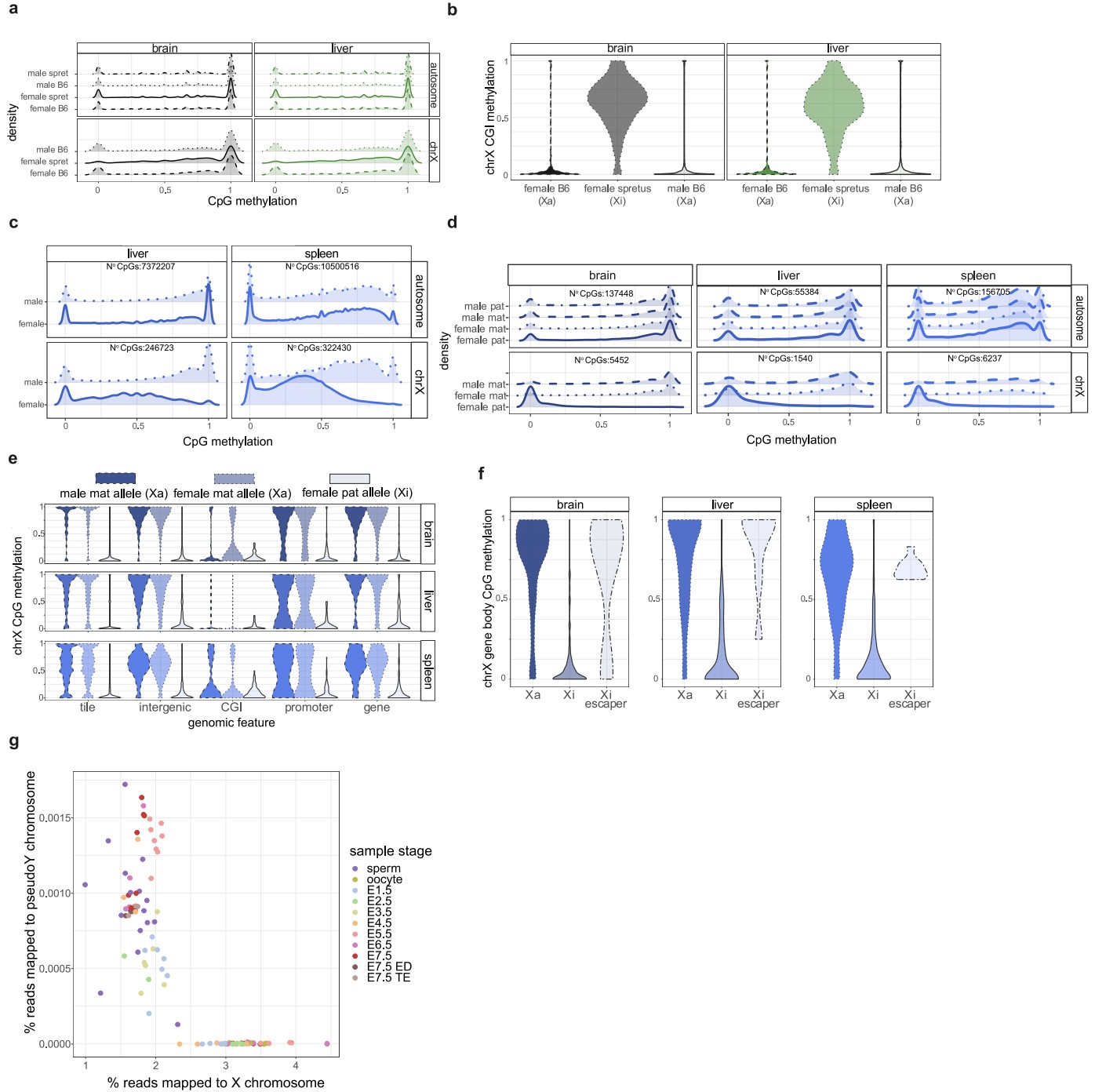

**Extended Data Fig. 5 | DNA methylation status of the opossum X chromosome cont. a** Allele-specific methylation distribution of the autosomes and X chromosome in adult mouse male and female brain and liver, represented as density plots showing the distribution of the data and the probability of a variable being a certain value. **b**. CGI methylation on the inactive and active X for data in **a. c**. Methylation distribution of the autosomes and X chromosome in adult opossum male and female liver. **d**. Allele-specific methylation analysis of the paternal and maternal alleles for data in **c**. **e**. Methylation at specific genomic features on the inactive and active X in adult opossum brain, liver and spleen. **f**. Gene body methylation on the inactive and active X for genes subject to or escaping XCI for female samples in **e. g**. Sexing of gamete and embryo samples via read mapping to the X and Y chromosome.

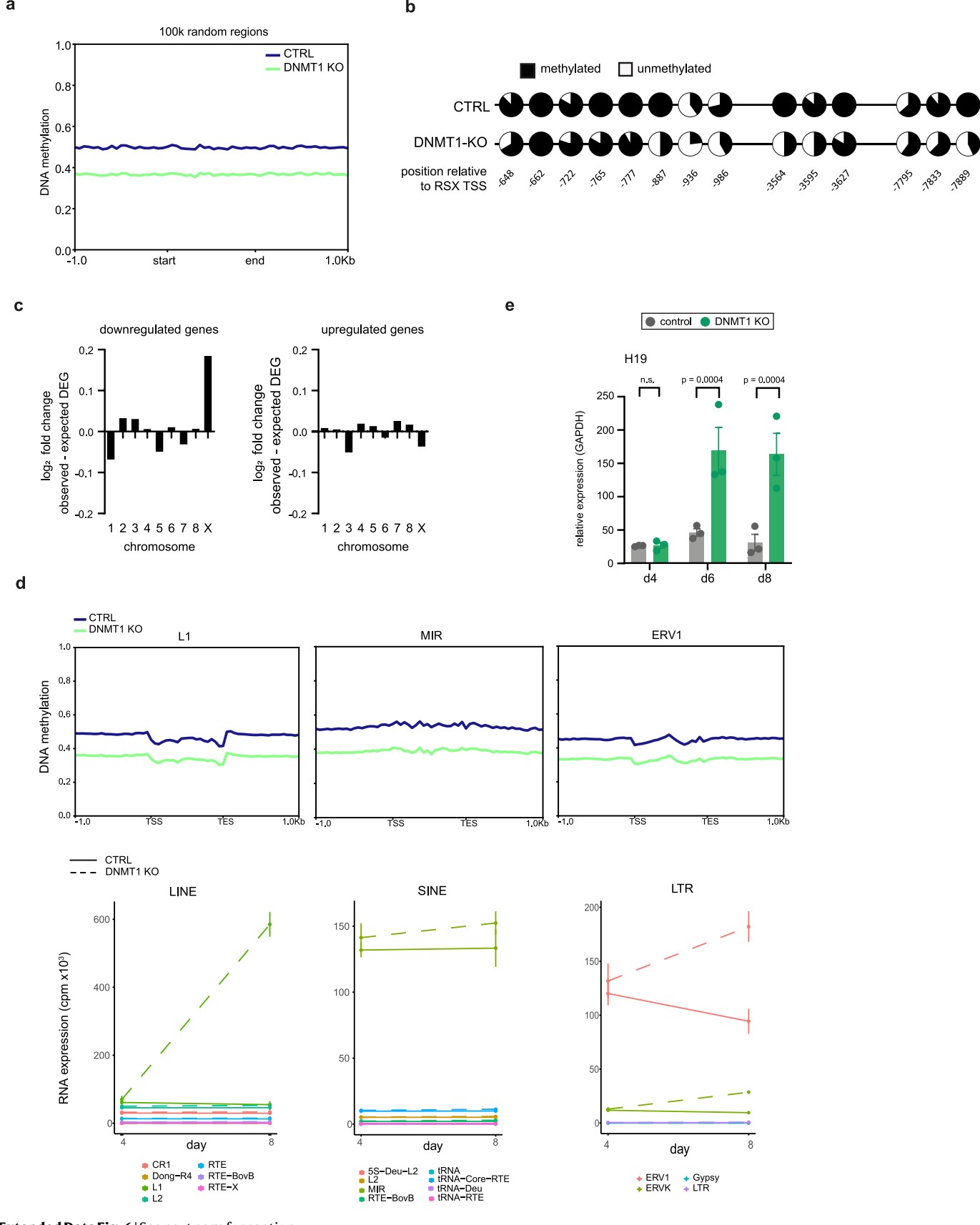

**Extended Data Fig. 6 |** See next page for caption.

**Extended Data Fig. 6 | Deletion of *DNMT1* in opossum immortalised male fibroblasts. a**. Metaplot of 100,000 randomly selected 1000 nucleotide regions demonstrating global decrease in DNA methylation in *DNMT1*-deleted fibroblasts at day 4. **b**. Mosaic loss of DNA methylation at the *RSX* promoter post-*DNMT1* deletion at day 4. **c**. Proportion of upregulated and downregulated genes by chromosome in *DNMT1*-deleted fibroblasts at day 8. **d**. Methylation and expression of repetitive elements following *DNMT1*-deletion. Above, metaplots showing decreased DNA methylation at L1, MIR and ERV1 repetitive elements 4 days after *DNMT1*-deletion. Below, line graphs showing transcriptional de-repression of L1, MIR, ERV1 and ERVK elements by day 8 after *DNMT1*-deletion. $N_{control} = 3$, $N_{DNMT1KO} = 3$. Error bars = 1.96*SE. Each point represents the mean of the replicates. **e**. qPCR analysis of *H19* in *DNMT1* deletant fibroblasts. $N_{control} = 3$, $N_{DNMT1KO} = 3$. Unpaired t-test. Error bars = SEM. Each point represents the mean of the replicate.

# Reporting Summary

## Statistics

For all statistical analyses, confirm that the following items are present in the figure legend, table legend, main text, or Methods section.

| n/a | Confirmed | |
|---|---|---|
| ☐ | ☒ | The exact sample size (*n*) for each experimental group/condition, given as a discrete number and unit of measurement |
| ☒ | ☐ | A statement on whether measurements were taken from distinct samples or whether the same sample was measured repeatedly |
| ☐ | ☒ | The statistical test(s) used AND whether they are one- or two-sided<br>*Only common tests should be described solely by name; describe more complex techniques in the Methods section.* |
| ☒ | ☐ | A description of all covariates tested |
| ☒ | ☐ | A description of any assumptions or corrections, such as tests of normality and adjustment for multiple comparisons |
| ☐ | ☒ | A full description of the statistical parameters including central tendency (e.g. means) or other basic estimates (e.g. regression coefficient) AND variation (e.g. standard deviation) or associated estimates of uncertainty (e.g. confidence intervals) |
| ☐ | ☒ | For null hypothesis testing, the test statistic (e.g. *F*, *t*, *r*) with confidence intervals, effect sizes, degrees of freedom and *P* value noted<br>*Give P values as exact values whenever suitable.* |
| ☒ | ☐ | For Bayesian analysis, information on the choice of priors and Markov chain Monte Carlo settings |
| ☒ | ☐ | For hierarchical and complex designs, identification of the appropriate level for tests and full reporting of outcomes |
| ☒ | ☐ | Estimates of effect sizes (e.g. Cohen's *d*, Pearson's *r*), indicating how they were calculated |

*Our web collection on statistics for biologists contains articles on many of the points above.*

## Software and code

Policy information about availability of computer code

| Data collection | No software was used. |
|---|---|
| Data analysis | Code used in preparation and analysis of data and the generation of figures is available at github.com/bleeke.<br>Software used:<br>FastQC 0.11.5<br>TrimGalore 0.6.0<br>SAMTools 1.4<br>SNPsplit 0.3.4<br>BWA-MEM<br>BCFTools<br>Varscan<br>GATK<br>BEDtools<br>Fiji/ImageJ 2.0.0<br>Bismark 0.18.0<br>HISAT2 2.1.0<br>telescope 1.0.3<br>Nextflow/23.10.0<br>Nextflow/21.04.0<br>Singularity/3.6.4<br>Deeptools |

```
RNA-seq nf-core pipeline v3.2 and 3.12
methylseq nf-core pipeline 2.5.0
R 3.6.0 and 4.2.2
R packages
ggplot2 3.2.0
methylKit 1.4.1
genomicRanges 1.30.3
DESeq2 1.36
Rsubread 1.28.1
scater 1.14.6
Seurat 4.3.0
```

For manuscripts utilizing custom algorithms or software that are central to the research but not yet described in published literature, software must be made available to editors and reviewers. We strongly encourage code deposition in a community repository (e.g. GitHub). See the Nature Portfolio guidelines for submitting code & software for further information.

## Data

Policy information about availability of data

All manuscripts must include a data availability statement. This statement should provide the following information, where applicable:

- Accession codes, unique identifiers, or web links for publicly available datasets
- A description of any restrictions on data availability
- For clinical datasets or third party data, please ensure that the statement adheres to our policy

BS-seq and RNA-seq data have been deposited at GE (accession number GSE206499). WGS data has been deposited at SRA (accession number PRJNA819000). Additional publicly available datasets used in this paper are accessible at GEO (GSE163620, GSE71434, GSE101571, GSE163620 and GSE71985), DDBJ (DRA006642 and DRA000570), and ArrayExpress (E-MTAB-7515). Reference genomes mm10, MonDom5 and ASM229v1 were accessed from Ensembl (https://www.ensembl.org/index.html) and mouse strain variants from the Mouse Genomes Project (https://www.sanger.ac.uk/data/mouse-genomes-project/)

## Research involving human participants, their data, or biological material

Policy information about studies with human participants or human data. See also policy information about sex, gender (identity/presentation), and sexual orientation and race, ethnicity and racism.

| Reporting on sex and gender | NA |
|---|---|
| Reporting on race, ethnicity, or other socially relevant groupings | NA |
| Population characteristics | NA |
| Recruitment | NA |
| Ethics oversight | NA |

Note that full information on the approval of the study protocol must also be provided in the manuscript.

# Field-specific reporting

Please select the one below that is the best fit for your research. If you are not sure, read the appropriate sections before making your selection.

☒ Life sciences        ☐ Behavioural & social sciences        ☐ Ecological, evolutionary & environmental sciences

For a reference copy of the document with all sections, see nature.com/documents/nr-reporting-summary-flat.pdf

# Life sciences study design

All studies must disclose on these points even when the disclosure is negative.

| Sample size | Sample sizes were not predetermined for opossum embryo BS-seq or scNMT-seq. The number of samples for each timepoint was dictated by litter size upon collection. Where sample availability permitted, for key timepoints we included multiple litters from separate collections. We ensured a minimum of 3 embryos of each sex at each timepoint. Our analysis of genomic coverage level at different genomic features supports our low input rare sample approach, as does our replication of previously reported mouse blastocyst methylation levels from similar low-input preparations. For opossum adult BS-seq, we collected 3 male and 3 female samples per tissue. For mouse adult BS-seq, we collected 3 male and 3 female samples per tissue. Mouse sperm and blastocyst BS-seq: sperm n=2 libraries of ~100 , E3.5 embryos n= 3. |
|---|---|

| Data exclusions | Some low-input BS-seq samples failed to amplify libraries and were therefore not further analysed. These are therefore not included in the final n number. |
|---|---|
| Replication | For opossum BS-seq we did not replicate over and above the initial number of collected samples due to limited availability of opossum embryo samples. Where sample availability permitted, for key timepoints we included multiple litters from separate collections and library preparations, and after QC in silico grouped all samples from the same timepoint for further analysis.  We included mouse embryo, sperm, and brain BS-seq samples as a replication of prior published mouse BS-seq experiments as a validation of the method. For opossum scNMT-seq we collected > 200 single cells, resulting in >50 cells per timepoint/lineage, and analysed the single cells separately, showing the range of the data across all single cells. |
| Randomization | There were no experimental groups as this was a wild-type time-course study. |
| Blinding | There were no group allocations in this study (as explained in 'Randomization' above). |

# Reporting for specific materials, systems and methods

We require information from authors about some types of materials, experimental systems and methods used in many studies. Here, indicate whether each material, system or method listed is relevant to your study. If you are not sure if a list item applies to your research, read the appropriate section before selecting a response.

## Materials & experimental systems

| n/a | Involved in the study |
|---|---|
| ☐ | ☒ Antibodies |
| ☐ | ☒ Eukaryotic cell lines |
| ☒ | ☐ Palaeontology and archaeology |
| ☐ | ☒ Animals and other organisms |
| ☒ | ☐ Clinical data |
| ☒ | ☐ Dual use research of concern |
| ☒ | ☐ Plants |

## Methods

| n/a | Involved in the study |
|---|---|
| ☒ | ☐ ChIP-seq |
| ☒ | ☐ Flow cytometry |
| ☒ | ☐ MRI-based neuroimaging |

## Antibodies

| Antibodies used | #75-268, Anti 5hMeC, NeuroMab (1:1000)<br>#07-442, H3K9me3, MerckMillipore (1:200)<br>#ab1791, H3, Abcam, (1:100)<br>#BI-MECY-0100, 5MeC, Eurogentec (1:100) |
|---|---|
| Validation | We validated the use of the above antibodies by reproducing staining patterns previously published in mouse embryos (Extended Data Figure 2b), before applying them to opossum embryos. |

## Eukaryotic cell lines

Policy information about cell lines and Sex and Gender in Research

| Cell line source(s) | Fibroblasts were derived from a male opossum neonate from our colony and immortalised by SV40-tag virus infection. |
|---|---|
| Authentication | Cell lines were not authenticated as they were derived by us. |
| Mycoplasma contamination | Cells were routinely tested for mycoplasma infection and found to be negative. |
| Commonly misidentified lines<br>(See ICLAC register) | No commonly misidentified cell lines were used in this study. |

## Animals and other research organisms

Policy information about studies involving animals; ARRIVE guidelines recommended for reporting animal research, and Sex and Gender in Research

| Laboratory animals | Grey short-tailed opossum: Monodelphis domestica gametes, embryos from embryonic day 1.5- 7.5, adult (> 6 months < 2 years). Mice: Mus musculus and Mus musculus x Mus spretus F1 cross. Embryos from embryonic day 3.5, adults (> 2 months < 6 months). |
|---|---|
| Wild animals | No wild animals were used. |
| Reporting on sex | For opossum embryos, we collected all embryos from each litter, but ensured a minimum of three embryos of each sex. Sex was |

| Reporting on sex | inferred based on ratio of reads mapping the X and pseudoY chromosome (Extended Data Figure 5g). For adult experiments, we collected 3 male and 3 female opossums and sex was identified based on physical sex characteristics. |
| Field-collected samples | No samples were collected from the field. |
| Ethics oversight | Opossums and mice were maintained in the Francis Crick Institute Biological Research Facility in accordance with United Kingdom Animal Scientific Procedures Act 1986 regulations (Project Licence P8ECF28D9) and subject to Francis Crick Institute ethical review. Additional opossums were housed at the University of Texas Rio Grande Valley under IACUC protocol AUP-19-31. |

Note that full information on the approval of the study protocol must also be provided in the manuscript.

## Plants

| Seed stocks | NA |
| Novel plant genotypes | NA |
| Authentication | NA |

