## [Peer Review File · Nature]

Evolutionary diversity in mammalian embryo DNA methylation reprogramming

Corresponding Author: Dr James Turner

Version 0:

Reviewer comments:

Referee #1

(Remarks to the Author)

The authors made detailed analysis of DNA methylation kinetics during early pre- and postimplantation embryos in opossum for the first time. They found that the opossum genome stays hypermethylated during early preimplantation development. EGA takes place during this period with a hypermethylated genome. The embryonic disk manifested the hypermethylated state resembling not only embryos at earlier stages but also adult somatic cells, whereas the trophoctoderm showed hypomethylation. Preferential demethylation of the paternal genome was not observed in opossum. The authors also examined expression of transposons and found that L1, MIR, and ERV1 were activated during preimplantation development despite the globally hypermethylated state of the genome. They further compared the methylation state on the Xa and Xi as well as the X in sperm and oocytes and found that the Xi was hypomethylated like the Xi in adult female somatic cells and this hypomethylated state was acquired during preimplantation development. Finally, they examined potential role of DNA methylation on the regulation of RSX and suggest that differential methylation between sperm and oocytes may have instructive role in imprinted XCI.

This study revealed unprecedented DNA methylation kinetics during pre- and postimplantation development of opossum. Although the data of the preimplantation embryos were rather limited due to low input, they carried out appropriate evaluation if such data obtained from low depth sequencing data could be referred to as representative of the overall genome. I would say that the findings of this study might not have significantly advanced our understanding of the regulatory mechanisms of XCI in marsupials, this study would draw broad fields of researchers and the findings should be valuable for the future studies.

Major comments

The authors suggested that differential methylation at the RSX locus between sperm and oocytes might have an instructive role in imprinted expression of RSX and subsequent imprinted XCI in opossum. I agree the finding of this correlation is potentially interesting, given that DNA methylation at promoters and CGIs did not seem to play a prominent role in gene silencing on the Xi in opossum, the effect of such differential methylation might be questionable. I am wondering if such differential methylation is also observed at autosomal imprinted loci. What extent of correlation is observed between DNA methylation at promoters and CGIs and expression of tissue-specific gene expression or transposons in the first place? Addressing these questions would help evaluate the impact of differential methylation at RSX on imprinted XCI.

Minor points

They describe that at each timepoint, individual embryos are examined. I do not think that they examined only a single embryo at each timepoints. They should provide information about how many individual embryos they examined at each timepoints.

Lower panels of Figure 3b show the difference of DNA methylation between sperm and oocyte, but it is not clear which of them shows higher or lower methylation. Although roughly 5% of CGIs manifests higher methylation as shown in red, it is not clear to me which of sperm or oocyte is higher. According to the description in the text, the graphs seem to show the levels in oocytes relative to sperm, but should be clearly mentioned in the legends and figures. In the top panels, I do not get what density on the y-axis means. Neither in similar graphs showing the distribution of DNA methylation with density on the y-axis Fig. 3d, Fig. 4a, b, c., and some extended Figures. Please provide more detailed explanation.

Referee #2

(Remarks to the Author)

Mammalian development is characterized by extensive reprogramming of DNA methylation patterns twice during development, at the gametic to embryonic transition and again, at the somatic to gametic transition. Whether this process, which may limit the possibility of intergenerational inheritance in mammals, is conserved from eutherians to marsupials, is unknown. Using whole genome bisulfite sequencing and transcriptomics, the authors describe here the dynamics of DNA methylation across early development in a marsupial model, the opossum *monodelphis domestica*. Some of the findings are novel: 1- the marsupial genome is not subject to active and passive demethylation across marsupial pre-implantation development, 2- transposable element expression peaks at the time of embryonic genome activation, 3- the Xist-like RSX long non-coding RNA harbors maternally inherited DNA methylation at its promoter, potentially explaining constitutive imprinting X inactivation in marsupials. Furthermore, the study confirms with better and unbiased resolution previous observations: 1- the inactive X chromosome progressively loses DNA methylation after fertilization, as opposed to the rest of the genome, 2- the trophectoderm is less methylated than the embryonic part. The work is of quality and has required working on a difficult and not well established animal model, with rudimentary genomic annotation. Unfortunately, this study stays rather superficial: the authors do not fully exploit their DNA methylation data, they do not make in-depth use of their transcriptomic data and there is no chromatin mark profiling that may help explaining some of the findings. On a related note, it is surprising that the authors never refer to or compare their findings with the most comprehensive study to date, recently carried out on gametic and embryonic material from multiple eutherian species (Wei Xie group, Lu et al., 2021, DOI: 10.1126/sciadv.abi6178), including methylome, chromatin and RNA-seq profiling in mouse, rat, human, pig and cow.

I am listing here a few points where the authors could improve expand their work and/or focus:

- Oocyte DNA methylation landscape: the distribution of oocyte DNA methylation has been previously reported to be divergent among eutherian species. Rodent oocytes present transcription-dependent DNA methylation coinciding with gene bodies, porcine and bovine oocytes show additionally DNA methylation at non-transcribed regions, and human oocytes show intermediate states. Where does the monodelphis oocyte methylome stand compared to these species? Could the authors incorporate/ compare with the Lu et al. (2021) datasets? Also, while the authors performed bulk RNA-seq in oocytes, they do not use these data to see how much transcription and DNA methylation correlate across the oocyte genome in marsupials. Moreover, there is no analysis of non-CG methylation, which is known to be abundant in eutherian oocytes. Finally, as in humans, it seems that DNMT3L is not significantly expressed in marsupial oocytes and may not be required to stimulate de novo DNA methylation activity in female gametes. This was not discussed.
- Autosomal genomic imprints: the manuscript focus on X-linked genomic imprinting (maternal differentially methylated region at the RSX promoter maintained post-fertilization), but there is no search into parental DMRs that would be associated with autosomal imprinting. It has been reported that fewer DNA methylation-dependent and life-long imprinted loci may exist in marsupials compared to eutherians. However these studies focused on a few candidate loci. Here, the authors have in hands DNA methylation patterns from the oocyte and the spermatozoon that could allow them to identify gametic DMRs, and track their fate across preimplantation and postimplantation development, potentially identifying life-long imprinted DMRs but also transient forms of parental imprinting, limited to early stages of development but not maintained to adulthood. Because there seems to be minimal DNA methylation changes after fertilization, parentally inherited differences in allelic DNA methylation should be prevalent. Unless the oocyte and the sperm methylome show convergent DNA methylation patterns in marsupials? However although genome-wide DNA methylation levels are not as drastically contrasted as in eutherians, the authors report higher DNA methylation at CGIs in oocytes compared to sperm in marsupials, but lower DNA methylation in oocytes compared to sperm at intergenic regions. So there should potentially many gametic DMRs to be called. Then, where do they locate? In gene bodies for the maternally methylated ones? In intergenic regions for paternally methylated ones?
- Maternal imprint at the RSX promoter: Post-fertilization maintenance of parental DNA methylation imprints strongly relies on DNA methylation-sensitive KRAB-ZFPs in eutherians, ZFP57 and ZFP445 to name them. Are these genes/proteins conserved in the monodelphis genome? And could the authors identify a ZFP57 binding motif within the RSX DMR?
- Transposable element (TE) expression: the authors report that despite maintaining DNA methylation across preimplantation development, several TE families are nonetheless up-regulated at the EGA stage in monodelphis embryos. However, only global DNA methylation levels were scored, across elements of a same family and across the length of these elements. In fact, a decrease of DNA methylation at TE promoters (where DNA methylation matters for their expression) would not be measurable by this analysis. The authors should report DNA methylation levels along TE elements (either by mapping reads on consensus sequences or using metaplot analysis of single mappers).
- Trophectoderm DNA methylation landscape: there is a large breadth of data regarding the DNA methylation landscape of extra-embryonic tissues in the mouse, from the A. Meissner or G. Kelsey lab. Notably, are there also partially methylated domains (PMDs) in marsupial trophectoderm too?

Referee #3

(Remarks to the Author)

This study on epigenetic reprogramming during early marsupial development is of interest because little is known about how and if it occurs, which is essential for insight into mammalian evolution. Much is known about similar events in eutherian mammals. The study suggests that DNA demethylation, a feature of early eutherian development, does not occur in a metatherian marsupial mammal. A previous study by this group showed differences between marsupials and eutherian mammals regarding the mechanism of X inactivation. For example, marsupials lack the XIST gene; instead, they have RSX, which functions similarly to XIST in mice and humans. This study provides additional information on how the paternal X

inactivation is regulated in marsupials.

This paper focuses on DNA methylation dynamics in embryos of a marsupial. The key data is summarised in Figure 1 (e.g. Fig 1c). Clearly, up to E4.5, there is little detectable demethylation, which contrasts with the early DNA demethylation in mouse embryos, starting with rapid DNA demethylation of the paternal genome and gradually in the maternal genome, reaching overall low levels in blastocysts. An indication of the approximate numbers of cells at different stages of marsupial development would help compare the available information in mice.

Looking at the data, whole embryos at E5.5 and E 6.5 appear to show some decline in DNA methylation. At E7.5, the authors record higher DNA methylation in the embryonic disc (ED) compared to the trophectoderm (TE). The authors believe that their observed decline in methylation is confined to TE. However, comparing the data for E6.5 and E7.5, is it equally possible that cells that give rise to ED undergo remethylation of the DNA? Analysis of the expression of some of the factors involved in DNA methylation, e.g. DNMT1B and UHRF1 shows a decline consistent with DNA demethylation (Fig1d). Is it possible to determine if there are differences in the expression of these factors in TE and ED, which might provide a starting point to explain the differences in DNA methylation in the two embryonic tissues?

The authors also claim that TE's relatively low DNA methylation levels are a "conserved feature" in marsupials and mice. For a definitive conclusion regarding conservation, the authors need information on the underlying mechanisms and what accounts for the differences in DNA methylation between DE and TE in marsupial embryos. Are there mechanisms that protect this modification in the DE, or do factors actively promote demethylation in TE? The authors should also consider the established differences in the development of blastocysts in marsupials and mice for their interpretations; for example, both the DE and TE tissues originate from unilaminar marsupial blastocysts.

The Discussion reveals an interpretation of the data, for example, the authors suggest that the "evolutionary arms race" (see line 215) between parental genomes in mice is absent in marsupials, based on Haig's parental conflict hypothesis, but now there are other interpretations. There is no evidence that the maternal oocyte cytoplasm 'strips paternal imprints.....'. Perhaps the DNA of parental pronuclei in marsupials do not inherit differential DNA methylation. Other statements also need justification. What is known is that the imprints in mice and humans are protected by specific Krab zinc finger proteins (KZNF), which are absent in marsupials, but some believe that there might be alternatives in marsupials. Some of the older literature cited is superseded; for example, "Our data thus support the hypothesis that evolutionary acquisition of Dppa3 accompanied the development of embryonic DNA demethylation⁵⁸, citing one paper but does not take into account other studies reaching different conclusions. Some further data on precisely what happens to DNA methylation in the embryo, ED and TE would help to reach appropriate conclusions.

Version 2:

Reviewer comments:

Referee #1

(Remarks to the Author)

The authors have done many additional experiments to address the criticisms that others and I have raised. They have increased the number of embryos analyzed by BS-seq to compensate for the lower input. They also performed a detailed analysis of the methylation status of potential imprinted genes. In addition, they addressed the role of DNA methylation in the regulation of RSX by knocking out DNMT1. I think that all these experiments basically strengthen their claim that DNA methylation plays an important role in imprinted XCI and genomic imprinting in marsupials. Finally, I would like the authors to consider addressing the following points.

RSX RNA was expressed and coated the X chromosome upon disruption of DNMT1 in male opossum fibroblasts. I think that the author should show the effect of knocking out three paralogs of DNMT1 on the global DNA methylation in immortalized male fibroblasts. In addition, although they showed a partial loss of DNA methylation at the RSX promoter in Extended data Fig. 5h, this should be provided in parallel with the data of parental male fibroblasts, in which this region is expected to be hypermethylated. I am also wondering if this ectopically expressed RSX RNA caused silencing of X-linked genes. If they did not examine this, I suggest the authors carry out RNA-FISH for expression of X-linked genes in combination with RSX. It would also be interesting to see the effect of knocking out DNMT1 paralogs on the silencing status of transposons as well.

Referee #2

(Remarks to the Author)

The authors have provided satisfactory answers to my requests of additional analyses and discussion. I congratulate them for this very important piece of work that will advance the epigenetic field, at the intersection of development and evolution. There may be an error in the Discussion, page 8 line 241: "DNA METHYLATION is mild and transient in the epiblast but sustained in the trophectoderm". I guess it should be read instead: "DNA demethylation" or "DNA methylation loss".

Referee #3

(Remarks to the Author)

The revised version of the paper presents more accurate interpretations of the findings, with additional experiments to support the conclusions. It highlights the fundamental difference in the DNA methylation dynamics between opossums and

eutherian mammals during cleavage stages. The two primary lineages, embryo and the trophoblast in the opossum develop from a unilaminar blastocyst. In contrast, in the eutherians, the two lineages start to develop as separate entities during cleavage divisions while undergoing rapid DNA demethylation before the emergence of blastocysts. In the opossum, cells are hypermethylated during early cleavage divisions.

They add more mechanistic insights on the mechanism of X-inactivation in an in vitro experiment to elucidate the role of RSX. They also describe potential novel imprinted genes. Are these different from those described by Cao et al., (2023). The role of ZFP445 in safeguarding their inheritance will require further work. Cao et al. (2023) identified differentially methylated control elements for imprinted genes; could they be investigated for ZFP445 (which they claim has diverged from the Eutherian counterpart) for binding using the opossum fibroblast cultures?

Specific points:

Lines 38-39: There is a significant difference between the Metatherians and Eutherians that diverged 160Ma. The statement should reflect this.

Lines 55-62: The marsupial system of DNA demethylation is less pronounced and, therefore, unlike in the Eutherians.

Line 58: Are paternal imprints erased during early development? Please cite a supporting reference.

Line 63: Is anything known about the role, if any, of the TET enzymes, especially TET1?

Fig 1D: There is no size difference in the size of the pronuclei as in the Eutherians. Could this be because of rapid active DNA demethylation of the paternal pronucleus in the latter and chromatin decondensation?

Line 277: In the genetic experiment were both DNMT1A and DNMT1B mutated? Are they functionally similar? Would it be possible to use CRISPR to alter the epigenetic status of RSX for direct analysis of the functional impact?

(Remarks on code availability)

I have not checked the code

Version 3:

Reviewer comments:

Referee #1

(Remarks to the Author)

In response to the criticisms and comments I raised, the authors have conducted additional experiments concerning DNMT1 KO cells. The new data presented in the revised manuscript enhance the study and provide a compelling argument for the acceptance of their conclusions by the readers. This work would be of interest to readers in a broad field of research and I have no further comment.

Response to Reviewers: MS 2022-06-09763

We thank the Reviewers for their helpful comments. We have addressed all their points, and this has greatly enhanced our manuscript, and allowed us to make additional interesting discoveries. The revisions took a considerable amount of time, because our opossum colony is a rare resource, comprising a limited number of animals, and we needed to apply new technologies to achieve the reviewers' requests. A point-by-point response is shown below, but we first highlight some major additions:

1) **CRISPR experiments to determine the role of DNA methylation in imprinted X-inactivation.** This was in response to Reviewer #1, who asked whether the promoter DNA methylation we observed at *RSX* was important for its regulation. Our findings support the conclusion that DNA methylation regulates imprinted X-inactivation in marsupials.

2) **Sequencing of additional bulk samples from timepoints where our genome coverage was previously low.** This was in response to Reviewer #1, who mentioned that the input was low. The new data strengthens our conclusion that DNA methylation is retained during opossum cleavage. It has also allowed us to address several questions raised by Reviewer #2, including the DNA methylation landscape of oocytes and transposons, and the fate of gamete differentially methylated regions. Interestingly, this new analysis has allowed us to identify 78 candidate imprinted genes in marsupials, and to show that most of these are maternally imprinted, as observed in eutherians. This may also be of interest to Reviewer #3, who raised imprinting in their comments on our discussion.

3) **Single-cell methylation / transcription data at three stages of blastocyst development.** This was in response to Reviewer #3, who asked whether the gradual decrease in DNA methylation at the blastocyst stage was specific to the trophectoderm or affected all cells of the blastocyst. This new analysis demonstrates that demethylation is modest and transient in the epiblast and sustained in the trophectoderm. It also suggests that differences in DNA methylation between the trophectoderm and epiblast are regulated by modulating expression of DNA methylation enzymes.

4) **Extensive revision of results and figures.** This is a result of the many additional analyses and new experiments. In particular, we have replaced the original section entitled "*DNA methylation dynamics at key developmental stages and transposon expression*" with a new section called "*DNA methylation landscape of sperm and oocytes*" to incorporate the requests from Reviewer #2 concerning identification of differentially methylated regions, the relationship between DNA methylation and gene expression in the opossum oocyte, interspecies comparisons of the oocyte DNA methylation landscape, and analysis of non-CpG methylation.

5) **Extensive revision of discussion.** In response to comments of Reviewer #3, we have replaced the original discussion with one that is more focussed and insightful.

Please find our point-by-point response to the Reviewers below: Please note that the line citations below refer to the **untracked/edited version** of the revised manuscript.

Reviewer #1

General comments

The authors made detailed analysis of DNA methylation kinetics during early pre- and postimplantation embryos in opossum for the first time. They found that the opossum genome stays hypermethylated during early preimplantation development. EGA takes place during this period with a hypermethylated genome. The embryonic disk manifested the hypermethylated state resembling not only embryos at earlier stages but also adult somatic cells, whereas the trophoctoderm showed hypomethylation. Preferential demethylation of the paternal genome was not observed in opossum. The authors also examined expression of transposons and found that LI, MIR, and ERV1 were activated during preimplantation development despite the globally hypermethylated state of the genome. They further compared the methylation state on the Xa and Xi as well as the X in sperm and oocytes and found that the Xi was hypomethylated like the Xi in adult female somatic cells and this hypomethylated state was acquired during preimplantation development. Finally, they examined potential role of DNA methylation on the regulation of RSX and suggest that differential methylation between sperm and oocytes may have instructive role in imprinted XCI.

Response: We thank the reviewer for this excellent summary of our findings.

This study revealed unprecedented DNA methylation kinetics during pre- and postimplantation development of opossum. Although the data of the preimplantation embryos were rather limited due to low input, they carried out appropriate evaluation if such data obtained from low depth sequencing data could be referred to as representative of the overall genome. I would say that the findings of this study might not have significantly advanced our understanding of the regulatory mechanisms of XCI in marsupials, this study would draw broad fields of researchers and the findings should be valuable for the future studies.

Response: We thank the reviewer for acknowledging the unprecedented DNA methylation kinetics revealed by our work and the fact that it will draw broad fields of researchers. Before our study, epigenetic profiling had never been performed before in marsupial embryos, so we were missing any information about this important mammalian outgroup. Our finding that global DNA methylation erasure does not occur provides important insight into the evolution and role of this process in mammals. Regarding the point about low input, we have now improved this by sequencing additional samples. Regarding whether the study significantly advances our understanding of the regulatory mechanisms of XCI in marsupials, we now add a functional CRISPR-based experiment that strengthens our conclusion that the regulatory mechanism involves DNA methylation of the maternal RSX allele (see below for more details).

Specific comments

1) *The authors suggested that differential methylation at the RSX locus between sperm and oocytes might have an instructive role in imprinted expression of RSX and subsequent imprinted XCI in opossum. I agree the finding of this correlation is potentially interesting, given that DNA methylation at promoters and CGIs did not seem to play a prominent role in*

gene silencing on the Xi in opossum, the effect of such differential methylation might be questionable. I am wondering if such differential methylation is also observed at autosomal imprinted loci. What extent of correlation is observed between DNA methylation at promoters and CGIs and expression of tissue-specific gene expression or transposons in the first place? Addressing these questions would help evaluate the impact of differential methylation at RSX on imprinted XCI.

Response. The reviewer asks whether DNA methylation is observed at autosomal imprinted loci in marsupials, and whether the differential DNA methylation regulates *RSX* expression. Regarding the first point, thirteen autosomally imprinted genes have been identified in marsupials (PMID: 36721950), and many of these have been shown in previous studies to exhibit differential DNA methylation. For this reason, DNA methylation is likely to be important for gene regulation in marsupials. These DNA-methylated imprinted genes include some that are also imprinted in eutherians, e.g., *H19*, *Igf2R* and *Peg10* and those imprinted only in marsupials, e.g., *Npdc1*, *Pou5f3*, *Nkrfl*, *Zfp68*, and *Rwdd2a*. We now cite these manuscripts in our revision (lines 173 and 686). We have also now performed a screen to identify novel autosomally imprinted genes in marsupials. We identified 20,800 sperm-specific and 22,921 oocyte-specific DMRs (lines 164-175; new Figure 3a), and tracked their fate during embryogenesis, using intermediate methylation as an indicator of retention (new Figure 3b; SNPs are rare in our laboratory opossum colony, precluding a SNP approach to imprint discovery). Using this information, we have identified 78 candidate autosomally imprinted genes in marsupials (new Fig3c, Supplementary Table 2). Some of the published imprinted marsupial genes are not present in the opossum genome assembly, and therefore would not be captured by our analysis.

Regarding the second point, we have now performed a functional experiment, which indicates that DNA methylation is required for maternal *RSX* silencing. In male fibroblasts, which carry a maternal X chromosome, the *RSX* allele is DNA methylated and is silent. However, when we use CRISPR to deplete the maintenance DNA methyltransferase *DNMT1* in opossum XY fibroblasts, *RSX* expression is activated (lines 227-235; new Figure 4d), and male cells now exhibit *RSX* clouds (new Figure 4e). To functionally examine whether DNA methylation also regulates autosomal imprinting, we assayed expression of *H19* in our *DNMT1*-depleted XY fibroblasts. We see an increase in *H19* expression, consistent with DNA methylation regulating *H19* silencing (new Figure 4d). Based on these findings, we conclude that DNA methylation regulates gene expression in marsupials.

2) *They describe that at each timepoint, individual embryos are examined. I do not think that they examined only a single embryo at each timepoints. They should provide information about how many individual embryos they examined at each timepoints.*

Response. All the information on numbers of embryos is now added as a new tab in Supplementary Table 1, which is mentioned on line 71.

3) *Lower panels of Figure 3b show the difference of DNA methylation between sperm and oocyte, but it is not clear which of them shows higher or lower methylation. Although roughly 5% of CGIs manifests higher methylation as shown in red, it is not clear to me which of sperm or oocyte is higher. According to the description in the text, the graphs seem to show the levels in oocytes relative to sperm, but should be clearly mentioned in the legends and figures. In the top panels, I do not get what density on the y-axis means. Neither in similar graphs showing*

the distribution of DNA methylation with density on the y-axis Fig. 3d, Fig. 4a, b, c., and some extended Figures. Please provide more detailed explanation.

Response. These are now both clarified in the associated figure legends.

Reviewer #2

General comments

*Mammalian development is characterized by extensive reprogramming of DNA methylation patterns twice during development, at the gametic to embryonic transition and again, at the somatic to gametic transition. Whether this process, which may limit the possibility of intergenerational inheritance in mammals, is conserved from eutherians to marsupials, is unknown. Using whole genome bisulfite sequencing and transcriptomics, the authors describe here the dynamics of DNA methylation across early development in a marsupial model, the opossum *monodelphis domestica*. Some of the findings are novel: 1- the marsupial genome is not subject to active and passive demethylation across marsupial pre-implantation development, 2- transposable element expression peaks at the time of embryonic genome activation, 3- the Xist-like RSX long non-coding RNA harbors maternally inherited DNA methylation at its promoter, potentially explaining constitutive imprinting X inactivation in marsupials.*

Response: We thank the reviewer for this excellent summary of our findings.

Furthermore, the study confirms with better and unbiased resolution previous observations: 1- the inactive X chromosome progressively lose DNA methylation after fertilization, as opposed to the rest of the genome, 2- the trophectoderm is less methylated than the embryonic part. The work is of quality and has required working on a difficult and not well established animal model, with rudimentary genomic annotation.

Response: We thank the reviewer for their positive comments.

Unfortunately, this study stays rather superficial: the authors do not fully exploit their DNA methylation data, they do not make in-depth use of their transcriptomic data and there is no chromatin mark profiling that may help explaining some of the findings. On a related note, it is surprising that the authors never refer to or compare their findings with the most comprehensive study to date, recently carried out on gametic and embryonic material from multiple eutherian species (Wei Xie group, Lu et al., 2021, DOI: 10.1126/sciadv.abi6178), including methylome, chromatin and RNA-seq profiling in mouse, rat, human, pig and cow.

Response: In our extensive revision, we have now sequenced more samples, more fully exploited the methylation data, and added transcriptional analysis. We also compare our findings with the published work on eutherian mammals (see below for more details). Our colony is a rare resource of small size, so we cannot perform an extensive analysis of additional chromatin marks. However, we feel that this manuscript should focus on DNA methylation, because this analysis has never been performed in marsupials, and as such it reveals many new and interesting findings.

Specific comments

1) *I am listing here a few points where the authors could improve expand their work and/or focus: Oocyte DNA methylation landscape: the distribution of oocyte DNA methylation has been previously reported to be divergent among eutherian species. Rodent oocytes present transcription-dependent DNA methylation coinciding with gene bodies, porcine and bovine oocytes show additionally DNA methylation at non-transcribed regions, and human oocytes show intermediate states. Where does the monodelphis oocyte methylome stand compared to these species? Could the authors incorporate/ compare with the Lu et al. (2021) datasets? Also, while the authors performed bulk RNA-seq in oocytes, they do not use these data to see how much transcription and DNA methylation correlate across the oocyte genome in marsupials. Moreover, there is no analysis of non-CG methylation, which is known to be abundant in eutherian oocytes. Finally, as in humans, it seems that DNMT3L is not significantly expressed in marsupial oocytes and may not be required to stimulate de novo DNA methylation activity in female gametes. This was not discussed.*

Response. We have now addressed all these questions regarding the oocyte methylation landscape:

- **interspecies comparisons of oocyte methylation:** we have examined the relationship between transcription and methylation in opossum oocytes and have compared this data with that in Lu et al (PMID:34818044 lines 177-186). We reproduced Lu et al's findings that in mouse and rat methylation is predominantly at transcribed gene bodies, while in cow and human there is also methylation at non-transcribed regions (new Figure 3d). We find that opossums represent a more extreme scenario, with non-transcribed regions showing even higher methylation than cow and human (new Figure 3d). This finding further emphasises the interesting differences between the methylation landscape in eutherians and opossums. Please note in this analysis that we did not include published pig data (PMID:32600441). The pig sample only contained a single replicate with low coverage. As such, too few sites (<1%) passed our coverage filter of three, and we deemed the data unsuitable to compare against the other species.
- **analysis of non-CG methylation:** we show that non-CpG methylation is enriched in opossum oocytes (lines 188-189; new Extended Data Figure 4b), as it is in eutherian oocytes, indicating that this feature is conserved in therians.
- **discussion on DNMT3L expression:** we now add a sentence in which we highlight that *de novo* DNA methylation in the opossum oocyte is likely DNMT3L-independent (lines 186-187).

2) *Autosomal genomic imprints: the manuscript focus on X-linked genomic imprinting (maternal differentially methylated region at the RSX promoter maintained post-fertilization), but there is no search into parental DMRs that would be associated with autosomal imprinting. It has been reported that fewer DNA methylation-dependent and life-long imprinted loci may exist in marsupials compared to eutherians. However these studies focused on a few candidate loci. Here, the authors have in hands DNA methylation patterns from the oocyte and the spermatozoon that could allow them to identify gametic DMRs, and track their fate across preimplantation and postimplantation development, potentially identifying life-long imprinted DMRs but also transient forms of parental imprinting, limited to early stages of development*

but not maintained to adulthood. Because there seems to be minimal DNA methylation changes after fertilization, parentally inherited differences in allelic DNA methylation should be prevalent. Unless the oocyte and the sperm methylome show convergent DNA methylation patterns in marsupials? However although genome-wide DNA methylation levels are not as drastically contrasted as in eutherians, the authors report higher DNA methylation at CGIs in oocytes compared to sperm in marsupials, but lower DNA methylation in oocytes compared to sperm at intergenic regions. So there should potentially many gametic DMRs to be called. Then, were do they locate? In gene bodies for the maternally methylated ones? In intergenic regions for paternally methylated ones?

Response. We thank the reviewer for these insightful comments. By sequencing additional samples, we have increased our sequence coverage of oocytes and sperm and used this information to identify gametic DMRs and to track their fate during embryogenesis (lines 164-175). We identified 20,800 sperm-specific and 22,921 oocyte-specific DMRs. We find that as in eutherians, oocyte DMRs are relatively enriched in intragenic regions and CGIs, while sperm DMRs are enriched in intergenic regions (new Figure 3a). We have tracked the fate of these DMRs during embryogenesis (new Figure 3b), using intermediate methylation as an indicator of retention (SNPs are rare in our laboratory opossum colony, precluding a SNP approach to imprint discovery). Using this information, we have identified 78 candidate autosomally imprinted genes in marsupials (new Figure 3c, Supplementary Table 2). This list includes some published marsupial imprinted genes, supporting the veracity of our approach (some published imprinted marsupial genes are not present in the opossum genome assembly, and therefore would not be captured by our analysis).

3) Maternal imprint at the RSX promoter: Post-fertilization maintenance of parental DNA methylation imprints strongly relies on DNA methylation-sensitive KRAB-ZFPs in eutherians, ZFP57 and ZFP445 to name them. Are these genes/proteins conserved in the monodelphis genome? And could the authors identify a ZFP57 binding motif within the RSX DMR?

Response. A previous publication demonstrated that *ZFP57* is not conserved in marsupials (PMID:30602440), so a *ZFP57* opossum specific binding site analysis across *RSX* cannot be performed. We did however check for a *ZFP57* binding site using a human *ZFP57* motif. No *ZFP57* binding sites were detected, consistent with *ZFP57* not being present in marsupials. However, the same publication demonstrated that *ZFP445* is conserved in marsupials. Inspired by the reviewer, we determined whether a *ZFP445* binding site resides within the *RSX* DMR. Since a consensus binding site for *ZFP445* has never been generated, we built one by applying Multiple Em for Motif Elicitation (MEME) to published *ZFP445* human and mouse ChIP-seq data (PMID: 35787786: GSE115387). Of the top 10 most significant motifs, one overlapped between mouse and human. We then used Find Individual Motif Occurrences (FIMO) to determine whether this tentative *ZFP445* motif resides in the *RSX* promoter. No *ZFP445* binding site was detected. This may be either because a *ZFP445* binding site does not exist, or that we cannot find it because the marsupial consensus has diverged from the eutherian consensus. Indeed, marsupial *ZFP445* has diverged significantly from eutherian *ZFP445* (PMID:30602440). Nevertheless, we now mention in the discussion that only *ZFP445* is conserved in marsupials and may contribute to imprint maintenance in these mammals (lines 251-253).

4) *Transposable element (TE) expression: the authors report that despite maintaining DNA methylation across preimplantation development, several TE families are nonetheless up-regulated at the EGA stage in monodelphis embryos. However, only global DNA methylation levels were scored, across elements of a same family and across the length of these elements. In fact, a decrease of DNA methylation at TE promoters (where DNA methylation matters for their expression) would not be measurable by this analysis. The authors should report DNA methylation levels along TE elements (either by mapping reads on consensus sequences or using metaplot analysis of single mappers).*

Response. We thank the reviewer for this excellent suggestion. Of the two approaches, we opted for the metaplot analysis of single mappers, because the alternative, consensus-based approach was hampered by the large variation in TE element length in the opossum. We used deepTools to deconvolve different TE element families, and to map methylation upstream, downstream, and within the gene body of these elements. We observed no overt loss of DNA methylation in the promoter at these elements (although we did observe increased methylation at LINE promoters during blastocyst development). Instead, the overall methylation followed the genome-wide trend, i.e. high in the cleavage stages and decreasing during blastocyst development (lines 118-122, 744-751; new Extended Data Figure 3). There are two possible explanations for our findings. The first is that the expression observed by RNAseq derives from individual subfamily loci for which we did not capture sufficient BS-seq coverage to reveal locus-specific patterns. Alternatively, TE expression may be triggered independent of DNA demethylation. We discuss these points in a new Supplementary Discussion, which can be found at the end of the Materials and Methods.

5) *Trophectoderm DNA methylation landscape: there is a large breadth of data regarding the DNA methylation landscape of extra-embryonic tissues in the mouse, from the A. Meissner or G. Kelsey lab. Notably, are there also partially methylated domains (PMDs) in marsupial trophoctoderm too?*

Response. We have now performed a more detailed analysis of the opossum trophoctoderm methylation landscape (lines 140-142). We find that partially methylated domains are indeed conserved in the opossum (new Figure 2b).

Reviewer #3

General comments

This study on epigenetic reprogramming during early marsupial development is of interest because little is known about how and if it occurs, which is essential for insight into mammalian evolution. Much is known about similar events in eutherian mammals. The study suggests that DNA demethylation, a feature of early eutherian development, does not occur in a metatherian marsupial mammal. A previous study by this group showed differences between marsupials and eutherian mammals regarding the mechanism of X inactivation. For example, marsupials lack the XIST gene; instead, they have RSX, which functions similarly to XIST in mice and humans. This study provides additional information on how the paternal X inactivation is regulated in marsupials.

Response: We thank the reviewer for highlighting the interest of our work.

Specific comments

1) *This paper focuses on DNA methylation dynamics in embryos of a marsupial. The key data is summarised in Figure 1 (e.g. Fig 1c). Clearly, up to E4.5, there is little detectable demethylation, which contrasts with the early DNA demethylation in mouse embryos, starting with rapid DNA demethylation of the paternal genome and gradually in the maternal genome, reaching overall low levels in blastocysts. An indication of the approximate numbers of cells at different stages of marsupial development would help compare the available information in mice.*

Response. All the information on numbers of embryos is added as a new tab in Supplementary Table 1, which is mentioned on line 71.

2) *Looking at the data, whole embryos at E5.5 and E 6.5 appear to show some decline in DNA methylation. At E7.5, the authors record higher DNA methylation in the embryonic disc (ED) compared to the trophectoderm (TE). The authors believe that their observed decline in methylation is confined to TE. However, comparing the data for E6.5 and E7.5, is it equally possible that cells that give rise to ED undergo remethylation of the DNA?*

Response. We really appreciate the reviewer raising this alternative possibility. To address it, we have now performed single-cell DNA methylation / transcription analysis at three stages of blastocyst development: E5.5, 6.5 and 7.5. Consistent with the reviewer's alternative hypothesis, we do indeed see that between E5.5 and E6.5, DNA methylation levels show a modest decline in both the epiblast and the trophectoderm (lines 129-140; new Figure 2a). Remarkably, one day later, at E7.5, DNA methylation levels increase again in the epiblast but continue to decline in the trophectoderm.

3) *Analysis of the expression of some of the factors involved in DNA methylation, e.g. DNMT1B and UHRF1 shows a decline consistent with DNA demethylation (Fig1d). Is it possible to determine if there are differences in the expression of these factors in TE and ED, which might provide a starting point to explain the differences in DNA methylation in the two embryonic tissues?*

Response. The reviewer is correct that levels of enzymes involved in maintenance of DNA methylation, i.e., *DNMT1A*, *DNMT1B* and *UHRF1* decline during embryogenesis. We also observe a decrease in the *de novo* DNA methyltransferase *DNMT3B* during embryogenesis. In response to the reviewer's request, we have analysed expression of these factors in the trophectoderm and embryonic disc, as well as in newly acquired single cells derived from the trophectoderm and epiblast. Intriguingly, we find that expression of *DNMT1A*, *DNMT1B*, *UHRF1* and *DNMT3A/B* increase in the epiblast, coincident with the increase DNA methylation in this lineage. Conversely, levels of these enzymes fall in the trophectoderm, concomitant with further loss of DNA methylation in this lineage (lines 147-160; new Figure 2c and Extended Data Figure 4a). Based on this finding, we favour a model in which differential expression of the DNA methylation machinery contributes to the distinct methylation profiles in the trophectoderm and epiblast.

4) *The authors also claim that TE's relatively low DNA methylation levels are a "conserved feature" in marsupials and mice. For a definitive conclusion regarding conservation, the authors need information on the underlying mechanisms and what accounts for the differences*

in DNA methylation between DE and TE in marsupial embryos. Are there mechanisms that protect this modification in the DE, or do factors actively promote demethylation in TE?

Response. The possible mechanism underlying the differential methylation of the trophoctoderm and epiblast is discussed in our response to the previous (comment 3). Our data favour a model in which a decrease in expression of DNA methyltransferases promotes demethylation in the trophoctoderm.

5) *The authors should also consider the established differences in the development of blastocysts in marsupials and mice for their interpretations; for example, both the epiblast and trophoctoderm tissues originate from unilaminar marsupial blastocysts.*

Response. We have now included two sentences that address the relevance of differences in the marsupial and mouse embryo to our interpretations. The first is in the introduction: lines 55-62.

“Global erasure of DNA methylation is not observed in non-mammalian vertebrates, and is thought to serve a mammal-specific function, for example to permit early embryonic genome activation, erasure of paternal imprints and epimutations, formation of the trophoctoderm, establishment of pluripotency, or expression of transposons regulating embryo development. These embryonic milestones occur over a more protracted period in the marsupial, making it a useful alternative model to investigate the role of DNA demethylation in mammalian embryogenesis”

The second is in the discussion: lines 243-246.

“Differences in DNA methylation between the epiblast and trophoctoderm are presumably achieved through cell autonomous mechanisms, because the marsupial blastocyst is unilaminar. We propose that this is regulated by differential expression of the DNA methylation machinery”.

6) *The Discussion reveals an interpretation of the data, for example, the authors suggest that the "evolutionary arms race" (see line 215) between parental genomes in mice is absent in marsupials, based on Haig's parental conflict hypothesis, but now there are other interpretations. There is no evidence that the maternal oocyte cytoplasm 'strips paternal imprints.....'. Perhaps the DNA of parental pronuclei in marsupials do not inherit differential DNA methylation. Other statements also need justification. What is known is that the imprints in mice and humans are protected by specific Krab zinc finger proteins (KZNF), which are absent in marsupials, but some believe that there might be alternatives in marsupials.*

Response. We thank the reviewer for raising these points. We have now revised the imprinting part of the discussion, removing speculation on the evolution of imprinting, and instead focusing on our discovery of new candidate imprinted genes, and the possible role of Krab zinc finger proteins in imprinting maintenance (lines 251-253).

7) *Some of the older literature cited is superseded; for example, "Our data thus support the hypothesis that evolutionary acquisition of Dppa3 accompanied the development of embryonic DNA demethylation⁵⁸, citing one paper but does not take into account other studies reaching different conclusions. Some further data on precisely what happens to DNA methylation in the embryo, ED and TE would help to reach appropriate conclusions.*

Response. We agree that the reported roles of Dppa3 in DNA methylation reprogramming are complex and potentially paradoxical, and we now include a more comprehensive statement and citations to reflect this in the discussion (lines 264-268). We have now addressed what happens to DNA methylation in the embryo, ED and TE (see responses above).

Response to Reviewers: MS 2022-06-09763B

We thank the Reviewers for re-assessing our paper and providing their helpful comments. We provide point-by-point responses to all reviewer comments below.

Reviewer #1

The authors have done many additional experiments to address the criticisms that others and I have raised. They have increased the number of embryos analyzed by BS-seq to compensate for the lower input. They also performed a detailed analysis of the methylation status of potential imprinted genes. In addition, they addressed the role of DNA methylation in the regulation of RSX by knocking out DNMT1. I think that all these experiments basically strengthen their claim that DNA methylation plays an important role in imprinted XCI and genomic imprinting in marsupials.

Response: We thank the reviewer for this summary of our paper.

Finally, I would like the authors to consider addressing the following points.

RSX RNA was expressed and coated the X chromosome upon disruption of DNMT1 in male opossum fibroblasts. I think that the author should show the effect of knocking out three paralogs of DNMT1 on the global DNA methylation in immortalized male fibroblasts. In addition, although they showed a partial loss of DNA methylation at the RSX promoter in Extended data Fig. 5h, this should be provided in parallel with the data of parental male fibroblasts, in which this region is expected to be hypermethylated. I am also wondering if this ectopically expressed RSX RNA caused silencing of X-linked genes. If they did not examine this, I suggest the authors carry out RNA-FISH for expression of X-linked genes in combination with RSX. It would also be interesting to see the effect of knocking out DNMT1 paralogs on the silencing status of transposons as well.

Response: We thank the reviewer for these suggestions on how to improve and extend our analysis of the *DNMT1* deletion experiments. We have now performed genome-wide bisulfite-sequencing and RNA-seq analysis of the *DNMT1*-deleted male fibroblasts. Using the bisulfite-sequencing data, we now show that DNA methylation is reduced in *DNMT1*-deleted compared to control fibroblasts (line 240-242): both genome-wide (Extended Data Fig. 6a), and at the *RSX* promoter (Extended Data Fig. 6b). Using the RNA-seq data, we assessed X-gene silencing as a result of ectopic *RSX* expression by examining the X:autosome (X:A) expression ratio and fold-change of genes on different chromosomes (line 247-249, Fig. 4g, Extended Data Fig. 6c). Interestingly, these findings show that *RSX* expression leads to lower expression from the single X chromosome. We detect decreased DNA methylation at several repetitive element families following *DNMT1*-deletion, and using RNA-seq find their expression is increased (line 249-251, Extended Data Fig. 6d).

Please note that to make room for these new analyses, we have moved our analysis of the expression of *H19* following *DNMT1*-KO to Extended Data Fig 6e.

Reviewer #2

The authors have provided satisfactory answers to my requests of additional analyses and discussion. I congratulate them for this very important piece of work that will advance the epigenetic field, at the intersection of development and evolution.

There may be an error in the Discussion, page 8 line 241: "DNA METHYLATION is mild and transient in the epiblast but sustained in the trophoctoderm". I guess it should be read instead : "DNA demethylation" or "DNA methylation loss".

Response: We thank the reviewer for their positive comments. We have corrected the typo in the Discussion (line 264).

Reviewer #3:

The revised version of the paper presents more accurate interpretations of the findings, with additional experiments to support the conclusions. It highlights the fundamental difference in the DNA methylation dynamics between opossums and eutherian mammals during cleavage stages. The two primary lineages, embryo and the trophoblast in the opossum develop from a unilaminar blastocyst. In contrast, in the eutherians, the two lineages start to develop as separate entities during cleavage divisions while undergoing rapid DNA demethylation before the emergence of blastocysts. In the opossum, cells are hypermethylated during early cleavage divisions.

Response: We thank the reviewer for this summary of our work.

They add more mechanistic insights on the mechanism of X-inactivation in an in vitro experiment to elucidate the role of RSX. They also describe potential novel imprinted genes. Are these different from those described by Cao et al., (2023). The role of ZFP445 in safeguarding their inheritance will require further work. Cao et al. (2023) identified differentially methylated control elements for imprinted genes; could they be investigated for ZFP445 (which they claim has diverged from the Eutherian counterpart) for binding using the opossum fibroblast cultures?

Response: 3 of the genes we identify as putative imprinted loci are also identified as imprinted genes in Cao *et al* (line 187, Fig. 3c, Supplementary Table 2). We propose that the other loci we identify are candidate imprinted regions. The difference between the genes found in our study and Cao *et al.* may be due to methodology – we search for gamete differentially methylated regions (gDMRs) that are retained in adult tissues, whereas Cao *et al* search for allele-specific expression using RNA-seq.

The reviewer raises an interesting question regarding the potential role of ZFP445 in regulation of imprinting. Unfortunately, we cannot address this ChIP-based question, because marsupial and eutherian ZFP445 are significantly diverged, and the peptides recognised by commercially available eutherian anti-ZFP445 antibodies (Aviva Systems Biology #AVIVARP37533_P050 and Merck/Sigma-Aldrich #SAB2105378) are only 12% and 38% homologous to opossum ZFP445. Importantly, this experiment would not impact the main messages of this paper, which reveals differences in DNA methylation dynamics between eutherian and marsupial embryos.

Specific points:

Lines 38-39: There is a significant difference between the Metatherians and Eutherians that diverged 160Ma. The statement should reflect this.

Response: We thank the reviewer for pointing this out. We have amended this last line of the Abstract to highlight the divergence between marsupials and eutherians (line 40-41). Please note that the timing of divergence between marsupials and eutherians is mentioned in the opening line of the Introduction. Please also note that on re-reading the Abstract with fresh eyes, we reorganised the first sentence, placing the citations showing DNA demethylation in eutherian embryos at a more appropriate position within that sentence to improve clarity.

Lines 55--62: The marsupial system of DNA demethylation is less pronounced and, therefore, unlike in the Eutherians.

Response: With apologies, we do not understand this comment. Does the reviewer mean that data preceding our current study shows that DNA demethylation is less pronounced in marsupials? We have not been able to locate such information, but would be keen to add any relevant citations (for reference, the lines that the reviewer refers to currently read: “*Global erasure of DNA methylation is not observed in non-mammalian vertebrates^{15,16}, and is thought to serve a mammal-specific function, for example to permit early embryonic genome activation, erasure of paternal imprints and epimutations, formation of the trophoctoderm, establishment of pluripotency, or expression of transposons regulating embryo development⁷⁻¹⁰. These embryonic milestones occur over a more protracted period in the marsupial, making it a useful alternative model to investigate the role of DNA demethylation in mammalian embryogenesis*”).

Line 58: Are paternal imprints erased during early development? Please cite a supporting reference.

Response: We agree that the phrasing of this line was unclear, and have amended the Introduction to clarify that paternal methylation is erased during early development, rather than paternal imprints (line 68).

Line 63: Is anything known about the role, if any, of the TET enzymes, especially TET1?

Response: We have clarified in the Introduction that TET enzymes and *de novo* methyltransferases are conserved in opossums, in addition to the *DNMT1* maintenance methyltransferases (line 73). We present expression analysis of methylation regulating enzymes in Fig. 2c and Extended Data Fig. 4a.

Fig 1D: There is no size difference in the size of the pronuclei as in the Eutherians. Could this be because of rapid active DNA demethylation of the paternal pronucleus in the latter and chromatin decondensation?

Response: We thank the reviewer for raising this intriguing idea. However, in bovine zygotes there is active demethylation of the paternal genome, but no size difference between pronuclei (Dean *et al* 2001 PNAS). Therefore we cannot speculate about a link between opossum pronuclei size and methylation status.

Line 277: In the genetic experiment were both DNMT1A and DNMT1B mutated? Are they functionally similar? Would it be possible to use CRISPR to alter the epigenetic status of RSX for direct analysis of the functional impact?

Response: Opossum *DNMT1A* and *DNMT1B* sequences are highly similar, with the same protein domains annotated in both genes (Alvarez-Ponce *et al* 2018 PLoS ONE). Therefore, we used guide RNAs designed to target both genes simultaneously. This is now stated clearly in the Results (line 240-241). CRISPR-based DNA demethylation in marsupials has never been reported, and we have not yet managed to get this technique to work in our lab. Also, the *RSX* DMR is very large (approximately 8 kb), so it may be hard to remove even if the technology becomes available. We have added a statement to the Discussion to say that future studies should confirm the DNA methylation-mediated regulation of imprinted XCI using epigenome editing (line 277-279). We hope that the new data better characterising the *DNMT1* KO phenotype, including the finding of partial X silencing, is of interest to this reviewer.